# AlphaAgentEvo: Evolution-Oriented Alpha Mining via Self-Evolving Agentic Reinforcement Learning

**Ziyi Tang**[1]     **Xuexiong Yin**[1]     **Weixin Chen**[1]     **Zechuan Chen**[1]
**Yongsen Zheng**[2,3]     **Wenxuan Ye**[4]     **Keze Wang**[1*]     **Liang Lin**[1]

[1]School of Computer Science and Engineering, Sun Yat-Sen University
[2]College of Computing and Data Science, Nanyang Technological University, Singapore
[3]Digital Trust Centre, Nanyang Technological University, Singapore
[4]The Chinese University of Hong Kong
{tangzy27,yinxx7,chenwx228,chenzch6}@mail2.sysu.edu.cn,
yongsen.zheng@ntu.edu.sg, nbvincentelite@g.ucla.edu,
kezewang@gmail.com, linlng@mail.sysu.edu.cn

## Abstract

*Alpha mining* seeks to identify predictive alpha factors that generate excess returns relative to the market from a vast and noisy search space; however, existing *evolution*-based approaches struggle to facilitate the systematic *evolution* of *alphas*. Traditional methods, such as Genetic Programming (GP), cannot interpret natural language instructions and often fail to extract valuable insights from unsuccessful attempts, leading to low interpretability and inefficient exploration. Analogously, without mechanisms for systematic *evolution*, e.g., long-term planning and reflection, existing multi-agent approaches may easily fall into repetitive evolutionary routines, resulting in inefficient *evolution*. To overcome these limitations, we introduce **AlphaAgentEvo**, a self-evolving Agentic Reinforcement Learning (ARL) framework for *alpha mining*, which moves *alpha mining* beyond the brittle "search–backtest–restart" cycle toward a continuous trajectory of *evolution*. Guided by a hierarchical reward function, our agent engages in self-exploration of the search space, progressively learning basic requirements (e.g., valid tool calls) and then more complex objectives (e.g., continuous performance improvements). Through this process, the agent acquires advanced behaviors such as long-horizon planning and reflective reasoning, which enable it to actively react to the underlying state (e.g., market regime shifts) and realize a self-evolving agent, marking a step toward more principled and scalable *alpha mining*. Extensive experiments demonstrate that AlphaAgentEvo achieves more efficient *alpha evolution* and generates diverse and transferable *alphas*, consistently surpassing a wide range of baselines. Notably, with only 4B parameters, it outperforms LLM-driven evolution methods configured with state-of-the-art closed-source reasoning models, highlighting the promise of ARL for next-generation *alpha mining*.

## 1 Introduction

*Alpha mining*, which refers to uncovering quantitative signals that generate excess returns relative to the market, remains highly challenging due to its vast search space, high-variance feedback signals, spurious correlations, and high computational cost. These issues make the process both computationally intensive and prone to false discoveries, highlighting the need for more systematic approaches. In this context, the notion of *alpha evolution* becomes central: instead of viewing each candidate *alpha* as an independent trial, the process emphasizes progressively transforming an initial *alpha* into a superior version through multi-turn interactions that incorporate evaluative feedback, refine its structure, and enhance its performance. This evolution-oriented perspective not only increases the likelihood of uncovering genuinely effective *alpha factors* (hereafter referred to

---

*Corresponding author.

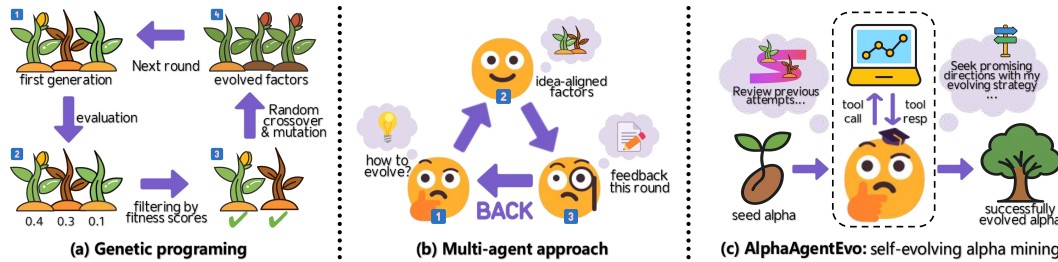

Figure 1: Comparison of evolution-oriented approaches for *alpha mining*.

as *alphas*) but also helps preserve their internal logic and interpretability across interaction turns, offering a principled alternative to ad hoc trial-and-error exploration. Despite this promise, existing approaches struggle to effectively realize such systematic *alpha evolution*.

Traditional evolution-oriented approaches, such as Genetic Programming (GP) (Lin et al., 2019; Schmidt & Lipson, 2010; Zhaofan et al., 2022), rely heavily on heuristic search and random mutation (Ren et al., 2024), without leveraging valuable insights from negative feedback, thereby missing opportunities to learn from failures and refine *alpha* design. Moreover, these methods cannot interpret their evolution process with human-readable language, which not only limits their usability but also increases the risk of generating *alphas* that capture spurious correlations (Shi et al., 2025b).

Emerging Large Language Models (LLMs) (Yang et al., 2025; DeepSeek-AI et al., 2025; OpenAI, 2025) and multi-agent frameworks (Tang et al., 2025; Li et al., 2025) accommodate more nuanced human instructions but often lack mechanisms for *self-evolution*, such as long-term planning and reflective reasoning from past outcomes. As a result, they tend to get trapped in repetitive local modifications, leading to inefficient exploration. Consequently, current alpha mining workflows remain myopic: they *search, backtest, and restart* rather than systematically evolve *alphas*. This gap calls for an evolution-oriented paradigm that couples deliberate planning with reflective reasoning to refine *alphas* over multi-turn trajectories.

To overcome these limitations, we introduce **AlphaAgentEvo**, the first self-evolving Agentic Reinforcement Learning (ARL) framework for *alpha mining*. First, AlphaAgentEvo aims to move *alpha mining* beyond the brittle "search–backtest–restart" cycle toward a continuous interaction trajectory, termed *evolution*, encouraging progressively refined *alphas* by searching diverse alphas grounded in a given seed. During the ARL process, an LLM-driven agent engages in large-scale exploration of the *alpha* search space, interacting with an evaluation tool under the guidance of a hierarchical reward structure. With this reward, the agent explicitly learns to satisfy basic requirements (e.g., constructing valid *alphas*) and then moves on to harder objectives (e.g., performance and streak improvements). More importantly, the agent acquires advanced behaviors, such as long-horizon planning and reflective reasoning, that enable it to actively react to the underlying state (e.g., market regime shifting). Ultimately, AlphaAgentEvo gives rise to a self-evolving agent, marking a step toward more principled and scalable *alpha mining*.

Results show that AlphaAgentEvo, at a relatively lightweight scale of 4B parameters, consistently outperforms GP, multi-agent, tool-use RL, and LLM-driven evolution baselines by a significant margin. Moreover, AlphaAgentEvo generates diverse and transferable *alphas*, avoiding over-exploitation of specific patterns and showing stronger out-of-sample robustness. These results underscore the superiority of our *self-evolving* ARL paradigm for evolution-oriented *alpha mining*.

Our key contributions are summarized as follows:

- We propose a novel framework, **AlphaAgentEvo**, that reformulates *alpha mining* from a brittle "search–backtest–restart" loop into an evolution-oriented paradigm. To the best of our knowledge, this is the **first** self-evolving agentic reinforcement learning framework for quantitative *alpha mining*.

- We design a hierarchical reward mechanism that guides an LLM-driven agent to perform multi-turn exploration of the *alpha* search space. This enables the agent to gradually acquire advanced behaviors, such as long-horizon planning and reflective reasoning, allowing it to adapt to changing market regimes and overcome performance bottlenecks in dynamic environments.

- Extensive experiments confirm that our method demonstrates strong evolution efficiency and validity, diversity, and transferability of the generated *alphas*, achieving strong generalization in the self-evolving agent. Remarkably, even with only 4B parameters, AlphaAgentEvo surpasses strong baselines powered by state-of-the-art LLMs on several metrics.

## 2 METHOD

### 2.1 PROBLEM DEFINITION

An *alpha factor* (or simply *alpha*) is a quantitative signal designed to predict future stock returns. We consider a stock universe $\mathcal{S} = \{s_1, \ldots, s_N\}$ over a time horizon $\mathcal{H} = \{h_1, \ldots, h_L\}$, with a feature matrix $\mathbf{X} \in \mathbb{R}^{N \times L \times d}$ where $d$ denotes the number of raw features. An *alpha* is a mapping $f : \mathbf{X}_h \mapsto r_{h+1}$, where $\mathbf{X}_h$ represents the market data observed up to and including time $h$ and $r_{h+1}$ is the subsequent return. Beyond static alpha mining, we view *alpha evolution* as *learning an evolution policy* instead of directly optimizing a single *alpha*. Let $\mathcal{D}_{\text{seed}}$ denote a distribution over expert-designed seed alphas. For a given seed $f_{\text{seed}} \sim \mathcal{D}_{\text{seed}}$, the policy $\pi$ interacts with the backtesting tool for $T$ turns and produces evolved *alphas* $\mathcal{F}_\pi(f_{\text{seed}})$. We evaluate each evolved factor on two market distributions: $\mathcal{D}_{\text{evo}}$, which corresponds to the in-distribution regimes observed during the agent's multi-turn evolution, and $\mathcal{D}_{\text{test}}$, which captures out-of-distribution market regimes. Formally, we learn the evolution policy by

$$\pi^\star = \arg\max_\pi \mathbb{E}_{f_{\text{seed}} \sim \mathcal{D}_{\text{seed}}} \left[ \max_{f \in \mathcal{F}_\pi(f_{\text{seed}})} \left( \mathbb{E}_{\mathbf{X} \sim \mathcal{D}_{\text{evo}}} s(f; \mathbf{X}) + \lambda \mathbb{E}_{\mathbf{X} \sim \mathcal{D}_{\text{test}}} s(f; \mathbf{X}) \right) \right] \tag{1}$$

$$\text{s.t. } \text{sim}(f, f_{\text{seed}}) \leq \delta \quad \text{for all } f \in \mathcal{F}_\pi(f_{\text{seed}}),$$

where $s(\cdot; \mathbf{X})$ is the performance scoring function on market features $\mathbf{X}$, $\lambda > 0$ trades off in-distribution fitness and out-of-distribution generalization, and $\text{sim}(\cdot, \cdot)$ is an AST-based structural similarity between alphas. The constraint $\text{sim}(f, f_{\text{seed}}) \leq \delta$ leads the policy $\pi$ to search in a local neighborhood of each seed, producing evolved alphas that are both stronger and still interpretable, rather than overfitting to noise via unconstrained global optimization.

### 2.2 SELF-EVOLVING AGENTIC REINFORCEMENT LEARNING

Existing RL-based fine-tuning approaches are typically designed for single-turn language modeling or reasoning, where evaluation is per response and cross-turn coupling is weak. In contrast, *alpha evolution* is inherently a multi-turn tool-in-the-loop process. To realize a self-evolving agent, we extend GRPO (Shao et al., 2024; DeepSeek-AI et al., 2025), into an Agentic Reinforcement Learning (ARL) formulation that directly optimizes the policy LLM in the *think-propose-evaluate* loop with an external evaluation tool $\mathcal{U}$.

In our formulation, each turn consists of policy-generated reasoning tokens and tool call tokens that trigger the tool, followed by tool response tokens (tool resp in Fig.2); all are appended to the trajectory, but only policy-generated tokens (indicated by $M_{i,t}$) contribute to gradients. To broaden exploration, at turn $t$, the policy LLM produces a set of $k_t$ parallel offspring $\mathcal{F}^{(t)}$ as candidates, which are jointly evaluated by $\mathcal{U}$. Note that when generating turn $t$'s tokens, the policy LLM conditions on the entire past trajectory $\tau_{1:t-1}$ to refine *alphas*, enabling reflective reasoning across previous attempts.

For each input $x$ sampled from the dataset $D$, a group of trajectories $\mathcal{T} = \{\tau_1, \ldots, \tau_G\}$ are rolled out by the old policy $\pi_{\text{old}}$. Their rewards are normalized within the group to estimate their relative advantages $\{\hat{A}_1, \ldots, \hat{A}_G\}$ using the group average reward as a baseline, i.e., $\hat{A}_g = \frac{R(\tau_g) - \mu_\mathcal{T}}{\sigma_\mathcal{T}}$, where $\mu_\mathcal{T}$ and $\sigma_\mathcal{T}$ denote the mean and standard deviation of $\{R(\tau_j)\}_{j=1}^G$. The optimization objective is

$$J_{\text{GRPO}}(\theta) = \mathbb{E}_{x \sim D, \, \mathcal{T} = \{\tau_1, \ldots, \tau_G\} \sim \pi_{\text{old}}} \left[ \frac{1}{G} \sum_{i=1}^G \frac{1}{\sum_t M_{i,t}} \sum_t M_{i,t} \, \min\left( \frac{\pi_\theta(\tau_{i,t} \mid x, \tau_{i,<t}, \mathcal{U})}{\pi_{\text{old}}(\tau_{i,t} \mid x, \tau_{i,<t}, \mathcal{U})} \, \hat{A}_{i,t}, \right. \right.$$

$$\left. \left. \text{clip}\left( \frac{\pi_\theta(\tau_{i,t} \mid x, \tau_{i,<t}, \mathcal{U})}{\pi_{\text{old}}(\tau_{i,t} \mid x, \tau_{i,<t}, \mathcal{U})}, \, 1 - \epsilon, \, 1 + \epsilon \right) \hat{A}_{i,t} \right) - \beta \mathbb{D}_{\text{KL}}[\pi_\theta \, \| \, \pi_{\text{ref}}] \right], \tag{2}$$

where $\tau_i$ is a complete evolution trajectory, $\tau_{i,t}$ its $t$-th token, and $\pi_\theta(\cdot) / \pi_{\text{old}}(\cdot)$ are the current/old policies conditioned on $x$, past tokens $\tau_{i,<t}$, and the tool $\mathcal{U}$. The mask $M_{i,t}$ excludes tool-emitted

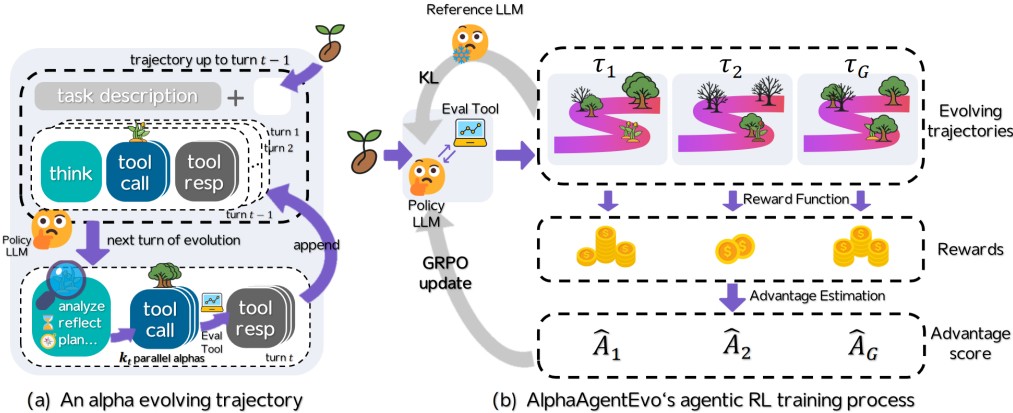

Figure 2: Overview of the AlphaAgentEvo framework. (a) An *alpha*-evolving trajectory. In each turn of the trajectory, the agent (policy LLM) generates multiple tool calls (i.e., *alpha* proposals) after analyzing and reflecting on previous *alphas* and their feedback (labeled as "tool resp") in the figure). (b) The multi-turn on-policy RL training process: each evolving trajectory is assigned a reward through the reward function. Trajectories that originate from the same seed *alpha* are grouped, and their rewards are jointly utilized for advantage estimation and policy LLM update.

tokens from gradients, $\frac{1}{\sum_t M_{i,t}}$ normalizes for effective length, clip($\cdot$) is ratio clipping with parameter $\epsilon$, and $\mathbb{D}_{\mathrm{KL}}$ is a KL penalty to a reference policy $\pi_{\mathrm{ref}}$ with weight $\beta$.

In summary, our ARL formulation adapts GRPO from single-turn text optimization to interactive, multi-turn *alpha evolution*. This enables the model to plan, analyze, and reflect throughout a long trajectory that progressively evolves *alphas* beyond the heuristic "search–backtest–restart" cycle.

## 2.3 ALPHA EVOLUTION REWARD FUNCTION

Unlike domains such as mathematics or generic tool use, *alpha mining* is central to quantitative investment but remains highly challenging due to its vast search space, noisy data with spurious correlations, and high computational cost, making it infeasible to rely on a single scalar reward. Moreover, the logical semantics of *alphas* and the need for efficient search are critical for effective alpha mining but rarely considered in other reasoning tasks. These challenges motivate a hierarchical reward function that enables principled *alpha evolution* throughout multiple turns, where AlphaAgentEvo integrates multiple objectives: ensuring valid *alpha* candidates, preventing excessive deviation from the seed *alpha*, promoting diversity exploration, rewarding performance improvements, and sustaining progress across turns.

Starting from an individual component, **Tool Call Reward** ($R_{\mathrm{tool}}$) provides feedback on correct tool usage and penalizes failed attempts, defined as $R_{\mathrm{tool}}(\tau) = \alpha_{\mathrm{succ}} \cdot N_{\mathrm{succ}} - \alpha_{\mathrm{fail}} \cdot N_{\mathrm{fail}}$, where $N_{\mathrm{succ}}$ and $N_{\mathrm{fail}}$ denote the number of successful and failed tool calls, respectively. Here, each $\alpha$ denotes the corresponding weighting coefficient (e.g., $\alpha_{\mathrm{succ}}$). Next, for direction-aware *alpha* generation, **Consistency Reward** ($R_{\mathrm{cons}}$) prevents excessive deviation from the seed *alpha* by penalizing candidates whose structural similarity $\mathrm{sim}(f_i, f_{\mathrm{seed}})$ falls below a lower threshold $h_{\mathrm{low}}$ (we set to 0.1 in our experiments), i.e., $R_{\mathrm{cons}}(\tau) = \sum_{f_i \in \mathcal{F}_{\mathrm{succ}}(\tau)} \alpha_{\mathrm{cons}} \cdot \mathbb{1}[\mathrm{sim}(f_i, f_{\mathrm{seed}}) > h_{\mathrm{low}}]$, where $\mathbb{1}[\cdot]$ is an indicator function. This serves as a soft constraint, preventing random drifts that may harm interpretability. **Exploration Reward** ($R_{\mathrm{expl}}$) encourages diversified exploration by rewarding *alphas* whose similarity to previously proposed ones remains low:

$$R_{\mathrm{expl}}(\tau) = \sum_{f_i \in \mathcal{F}_{\mathrm{succ}}(\tau)} \alpha_{\mathrm{exp}} \cdot \left(1 - \max_{f_j \in \mathcal{F}_{<i}(\tau)} \mathrm{sim}(f_i, f_j)\right). \tag{3}$$

where $\mathcal{F}_{\mathrm{succ}}(\tau)$ is the set of successfully runnable offspring *alphas* in trajectory $\tau$, $\mathcal{F}_{<i}(\tau)$ denotes all factors proposed before $f_i$ (including the seed). The structural similarity score $\mathrm{sim}(\cdot, \cdot)$ measured by Abstract Syntax Tree (AST) overlap (Tang et al., 2025) is written as:

$$\text{sim}(f_i, f_j) = \frac{|\text{AST}(f_i) \cap \text{AST}(f_j)|}{\max(|\text{AST}(f_i)|, |\text{AST}(f_j)|)}. \tag{4}$$

To encourage performance improvement while handling noisy metrics, **Performance Reward** ($R_{\textbf{perf}}$) uses a logarithmic scaling $\alpha_{\text{perf}} \cdot \log(1 + \exp(s(f^*) - \max(0, s(f_{\text{seed}}))))$. Finally, **Streak Reward** ($R_{\textbf{streak}}$) provides an additional bonus $\alpha_{\text{streak}} \cdot N_{\text{streak}}$, where $N_{\text{streak}}$ denotes the length of the longest sequence of progressive performance improvements within a trajectory, serving as a booster towards efficient *alpha evolution*.

Totally, the hierarchical reward of a trajectory $\tau$ is

$$R(\tau) = \frac{\min(R_{\text{cons}}(\tau), C_{\text{cons}}) + \min(R_{\text{expl}}(\tau), C_{\text{expl}})}{\min(R_{\text{tool}}(\tau), C_{\text{tool}})} + \min(R_{\text{perf}}(\tau), C_{\text{perf}}) \cdot \min(R_{\text{streak}}(\tau), C_{\text{streak}}), \tag{5}$$

where each component reward $R_j(\tau)$ is capped by its corresponding $C_j$ (e.g., $C_{\text{tool}}$) to avoid any single term from overwhelming the total reward. The tool-use term treats each tool call as a cost, preventing brute-force search through frequent tool calls and encouraging meaningful and efficient alpha evolution. This hierarchical reward structure transforms the sparse and noisy feedback from financial backtesting into dense, multi-dimensional signals. By balancing direction-aware consistency and exploration (normalized by tool usage) and integrating performance with sustained improvement through a multiplicative term, it progressively guides the agent from basic compliance to higher-level objectives, ultimately preventing collapse into repetitive patterns and enabling efficient *alpha evolution* (Wang et al., 2025).

## 3 EXPERIMENTS

In this section, we present the main experimental results, while **we strongly encourage readers to refer to Sec. E in the Appendix for a detailed case analysis** that provides a deeper understanding of the agent's *self-evolution* process. For details on the training statistics, please refer to Sec. C; for training configurations and the evaluation tool, please refer to Sec. D and Sec. F in the Appendix.

### 3.1 EXPERIMENT SETTINGS

**Datasets.** To systematically evaluate *alpha evolution* ability, we construct an expert-curated dataset, referred to as *AlphaEvo500*, which serves as an *alpha evolution* benchmark in this study. It consists of 350 seed *alphas* for training, 50 for validation, and 100 for testing, enabling a controlled yet diverse setting for evolutionary experiments. To further assess the generalization ability, we additionally incorporate *Alpha158* (Yang et al., 2020) as an extra test set.

**Backtesting settings.** Backtesting is conducted on the HS300 and CSI500 markets, spanning from January 2023 to November 2025, and covering both bearish and bullish market conditions. For model training, only one year of market data (2023-01-01 to 2024-01-01) is used to accelerate iterations. For evaluation, *alpha evolution* experiments are performed on two distinct periods: 2023-01-01 to 2024-01-01 (bearish) and 2024-01-01 to 2025-01-01 (bullish), with test split's *alphas* as the seeds. It should be noted that no data information is disclosed from the evaluation tool or the prompt, covering both the market and the time range. We adopt a single-factor evaluation protocol, where the cross-sectional values of each *alpha* are treated as signals, without extra processing. In each rebalancing period, we long at most the top 10% of stocks in the universe. The rebalancing frequency is set to every 5 trading days. Unless otherwise specified, other experimental settings are described at the beginning of each subsection. For detailed data and operators that we use in this paper, please see Sec. G in the Appendix.

**Evaluation metrics.** We evaluate the capability of AlphaAgentEvo to evolve *alphas* by computing the pass rate at the third and fifth turns, denoted as pass@3 and pass@5, respectively. For performance measurement, we adopt the Information Ratio (IR) as $s(f)$, which quantifies risk-adjusted excess return. Given a seed *alpha* $f_{\text{seed}}$ with score $s(f_{\text{seed}})$, **a generated *alpha* is regarded as successful if its score is higher than the seed and non-negative.** Formally, the pass rate at turn $T$ is defined as:

$$\text{pass@}T \;=\; \frac{1}{N}\sum_{j=1}^{N}\mathbb{1}\Big[\max_{f\in\bigcup_{t=1}^{T}\mathcal{F}^{(t)}} s(f) \;>\; \max\big(0,\, s(f_{\text{seed}}^{(j)})\big)\Big]. \qquad (6)$$

Here $N$ is the number of test cases, $\mathcal{F}^{(t)}$ is the set of evolved *alphas* at turn $t$, and the indicator $\mathbb{1}[\cdot]$ equals 1 if the success condition is satisfied and 0 otherwise. In addition, we report the *valid ratio* (VR), which measures the percentage of generated *alphas* that are syntactically valid and executable in backtesting, reflecting the reliability of the generation process.

For *alpha* performance measurement, we adopt the Annualized Excess Return (AER) to quantify the yearly excess investment return relative to the benchmark index and the Information Ratio (IR) to measure the risk-adjusted performance, which calculates the ratio between the AER and the annualized standard deviation of an *alpha*'s daily excess returns. With these evaluation metrics, we can comprehensively assess an *alpha's* profitability and risk-bearing capacity. We do not rely on cross-sectional correlation-based metrics (such as information coefficient or IC), since some of the seed *alphas* are designed as stock-selection signals in the form of Boolean expressions, where their values for unselected stocks are set to `NaN`, making these measures unreliable.

**Compared methods.** We compare our method against four categories of baselines: (i) genetic programming (GP) (Lin et al., 2019; Schmidt & Lipson, 2010; Zhaofan et al., 2022; Patil, 2023) with 4, 20, and 50 offspring per generation, representing traditional heuristic search; (ii) **LLM-driven evolutionary frameworks**, AlphaAgent (Tang et al., 2025) (multi-agent evolution) and GEPA (Agrawal et al., 2025) (reflective prompt evolution); (iii) a series of **reasoning LLMs** that evolve *alphas* via our unified pipeline, incorporating our base models (i.e., Qwen3-1.7B and Qwen3-4B-thinking (Yang et al., 2025)) and state-of-the-art models (GPT-5-mini (OpenAI, 2025), DeepSeek-R1 (Guo et al., 2025)); and (iv) a **tool-use RL approach** ToolRL (Qian et al., 2025). All methods utilize the same backtesting tool and adhere to identical tool call budgets for training and inference (4 offspring per turn, unless otherwise noted) to ensure a fair comparison. We also compare multi-factor portfolio performance with global optimization approaches (Zhu & Zhu, 2025; Fan & Shen, 2024; Ke et al., 2017) (see Appendix Sec. B).

## 3.2 Alpha Evolution Performance

To evaluate whether AlphaAgentEvo can consistently outperform existing evolution-oriented baselines in terms of *alpha evolution* capability, we first focus on VR and the pass rate, as illustrated previously. By evaluating on *AlphaEvo500* and *Alpha158 alpha* libraries across two periods, we further test the generalization ability of our approach under distinct market regimes. Note that AlphaAgentEvo models are trained with at most 3 turns.

Table 1: Performance comparison on *AlphaEvo500* across two markets during 2024-2025.

| Method | HS300 | | | CSI500 | | |
|---|---|---|---|---|---|---|
| | VR | Pass@3 | Pass@5 | VR | Pass@3 | Pass@5 |
| Qwen3-1.7B (Yang et al., 2025) | 0.676 | 0.08 | 0.11 | 0.657 | 0.35 | 0.43 |
| Qwen3-4B-thinking (Yang et al., 2025) | 0.942 | 0.36 | 0.47 | 0.951 | 0.68 | 0.78 |
| GPT-5-mini (OpenAI, 2025) | 0.970 | 0.75 | 0.88 | 0.972 | 0.73 | 0.82 |
| DeepSeek-R1 (Guo et al., 2025) | 0.872 | 0.68 | 0.71 | 0.886 | 0.71 | 0.86 |
| ToolRL-1.7B (Qian et al., 2025) | 0.864 | 0.74 | 0.78 | 0.851 | 0.66 | 0.74 |
| ToolRL-4B (Qian et al., 2025) | 0.954 | 0.75 | 0.81 | 0.961 | 0.73 | 0.76 |
| GEPA (Agrawal et al., 2025) (GPT-5-mini) | **0.992** | 0.87 | 0.90 | 0.971 | 0.86 | 0.91 |
| GEPA (Agrawal et al., 2025) (DeepSeek-R1) | 0.977 | 0.83 | 0.87 | **0.978** | 0.82 | 0.88 |
| AlphaAgentEvo-1.7B (*ours*) | 0.940 | 0.77 | 0.90 | 0.923 | 0.76 | 0.78 |
| AlphaAgentEvo-4B (*ours*) | 0.979 | **0.97** | **0.97** | 0.977 | **0.93** | **0.95** |

**Results on *AlphaEvo500*.** Table 1 shows the results in the HS300 and CSI500 markets. While GP's expression system is incompatible with *AlphaEvo500*, we are unable to test GP here. In terms of pass rates, Qwen3 and GPT-5-mini offer only limited improvements, and DeepSeek-R1 performs more strongly but inconsistently. For an agentic RL baseline, ToolRL, its pass@3 remains the same level as GPT-5-mini, but fails to generalize to a longer horizon due to the shortcomings in multi-turn planning. By contrast, AlphaAgentEvo achieves clear superiority: even the 1.7B version surpasses GPT-5-mini, while the 4B model outperforms the strongest baseline GEPA and attains

the best overall results. These findings demonstrate that our *self-evolving* agent not only effectively generalizes to different market regimes but also to longer evolution trajectories.

Table 2: Performance comparison on *Alpha158* across two periods.

| Method | 2023-01 – 2024-01 | | | 2024-01 – 2025-01 | | |
| --- | --- | --- | --- | --- | --- | --- |
| | VR | Pass@3 | Pass@5 | VR | Pass@3 | Pass@5 |
| GP (4 offspring) | 0.766 | 0.000 | 0.074 | 0.823 | 0.003 | 0.003 |
| GP (20 offspring) | 0.714 | 0.000 | 0.058 | 0.713 | 0.125 | 0.132 |
| GP (50 offspring) | 0.619 | 0.022 | 0.024 | 0.633 | 0.094 | 0.107 |
| AlphaAgent (Tang et al., 2025) (GPT-3.5-turbo) | 0.905 | 0.236 | 0.495 | 0.900 | 0.643 | 0.783 |
| AlphaAgent (Tang et al., 2025) (DeepSeek-R1) | 0.975 | 0.294 | 0.550 | 0.966 | 0.750 | 0.848 |
| Qwen3-1.7B (Yang et al., 2025) | 0.714 | 0.100 | 0.113 | 0.674 | 0.500 | 0.543 |
| Qwen3-4B-thinking (Yang et al., 2025) | 0.792 | 0.350 | 0.450 | 0.974 | 0.848 | 0.856 |
| DeepSeek-R1 (Guo et al., 2025) | 0.889 | 0.327 | 0.519 | 0.874 | 0.872 | 0.943 |
| GPT-5-mini (OpenAI, 2025) | **0.988** | 0.156 | 0.293 | 0.975 | 0.828 | 0.903 |
| AlphaAgentEvo-1.7B (*ours*) | 0.952 | 0.506 | 0.613 | 0.917 | 0.909 | 0.926 |
| AlphaAgentEvo-4B (*ours*) | 0.982 | **0.581** | **0.725** | **0.982** | **0.963** | **0.994** |

**Results on *Alpha158*.**   Table 2 reports the results on the external *Alpha158* library, which serves as an additional test set. GP reveals poor results with limited offspring size, while the 4-offspring setting is aligned with all other approaches. This highlights the inefficiency of purely heuristic search. The multi-agent framework AlphaAgent improves upon GP, yielding higher VR and reasonable pass rates, particularly with stronger backbone models. However, AlphaAgentEvo again achieves the most consistent improvements. AlphaAgentEvo-1.7B's VR remains above 0.91 across both market periods, while the 4B version shows a striking advantage in pass rates, with pass@5 exceeding 0.72 in the bearish period and reaching 0.994 in the bullish period, nearly saturating the success rate. This demonstrates not only superior evolutionary efficiency but also strong adaptability to different market styles and factor libraries.

**Summary.**   Across both datasets and market conditions, AlphaAgentEvo exhibits clear superiority over a wide range of strong baselines. Its ability to sustain both high valid ratios and high pass rates underscores the effectiveness of our *self-evolving* agent paradigm. Importantly, these results validate that the proposed framework not only accelerates the discovery of profitable *alphas* but also reduces invalid generations and enhances robustness under dynamic market environments.

## 3.3 EVOLUTION ANALYSIS.

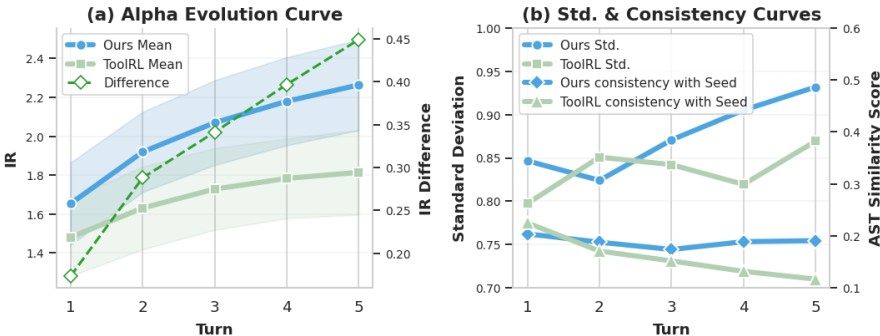

Figure 3: Comparison between AlphaAgentEvo and ToolRL's evolution trajectories.

To probe into how AlphaAgentEvo's agent and its generated *alphas* evolve across turns, in Fig. 3, we visualize its IR trajectory and per-turn standard deviation (Std.) along with average consistency score against ToolRL (Qian et al., 2025) on AlphaEvo500. For better visual clarity, we display the uncertainty band as ±0.25×std. In Fig. 3 (a), our approach's mean IR increases more rapidly than ToolRL, leading to a continually widening gap. Fig. 3 (b) shows that AlphaAgentEvo more aggressively explores in the search space as the agent continues to mine alphas, resulting in a higher standard deviation. Moreover, AlphaAgentEvo stays anchored to seed *alphas*, ensuring that improvements accumulate progressively rather than degenerating into uncontrolled search. These patterns cannot be explained by alpha-level evolution alone. The accelerating IR gains across turns, together with

the simultaneous rise in exploration and stabilized consistency, indicate that the agent's strategy in each turn is evolving (improving over time). These experience-dependent behaviors constitute clear evidence of agent-level self-evolution, rather than merely evolving individual *alphas*.

## 3.4 ABLATION STUDY

To verify the effectiveness of our ARL training and reward design, we compare validity rates before and after training and ablate two key reward components. As shown in Fig. 4(a), training markedly improves validity, confirming the model's ability to generate well-formed alphas. In Fig. 4(b)–(c), removing either the exploration or direction-aware reward lowers pass rates on both datasets, with the largest drops at pass@3 (AlphaEvo500: 0.65→0.54/0.51; Alpha158: 0.581→0.513/0.510). These results show that exploration and direction-awareness are both critical and complementary for efficient *alpha evolution*.

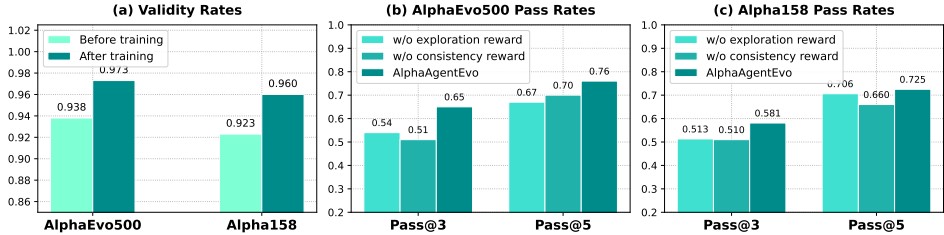

Figure 4: Ablation study on reward components.

## 3.5 DIVERSITY AND TRANSFERABILITY OF ALPHAS

To investigate whether models suffer from reward hacking (Wang et al., 2025) by overexploiting specific patterns, we analyze the structural similarity distribution of the top 20 generated *alphas* (evaluated on the *alpha evolution* period) using Eq. 4. Note that for GP, *Alpha158* serves as seeds due to an incompatible *alpha* calculation system; therefore, its results are provided solely for reference. For AlphaAgent, we reset its *alpha zoo* for each seed *alpha*.

In Fig. 5, the seed *alpha* library (*AlphaEvo500* test split) exhibits a broad similarity distribution, with an average pairwise similarity of 0.043 and a relatively high maximum similarity of 0.722, reflecting the presence of clusters of closely related *alphas* despite overall diversity. When comparing different models, our method achieves both a low average similarity (0.039) and a low maximum similarity (0.263), indicating that the generated *alphas* are more diverse. By contrast, models such as DeepSeek-R1 and Qwen3-4B tend to produce *alphas* with higher maximum similarity (0.583 and 0.600, respectively), suggesting partial over-concentration on specific patterns. AlphaAgent (GPT-4) also shows an elevated average similarity (0.058), suggesting that for different seed *alphas*, it may repeatedly fall into local optima with limited diversity. These statistics highlight a key advantage of our approach: **it does not overexploit narrow or spurious patterns and instead learns genuinely generalizable evolutionary strategies.**

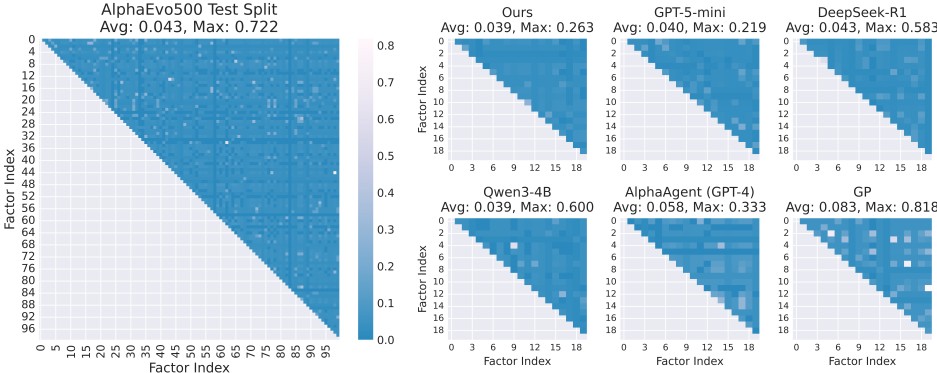

Figure 5: Similarity scores of top 20 *alphas* generated by different models.

To assess the out-of-sample performance of *alphas* from different LLMs, we collect two groups of evolved alphas from each LLM. While one group uses market data from 2023-01-01 to 2024-01-01 as an evolution period, the other uses 2024-01-01 to 2025-01-01 to evolve alphas. Then, the first group undergoes backtesting from 2024-01-01 to 2025-01-01 (test period 1), and the second group from 2025-01-01 to 2025-06-01 (until the datasets are created), noted as test period 2. These evolved *alphas* are sampled from each model's top-20 candidates without altering their original order **such that the selected subsets share the same average IR during the evolution period**. Specifically, the average evolution-period IR is 1.05 for test period 1 and 2.72 for test period 2.

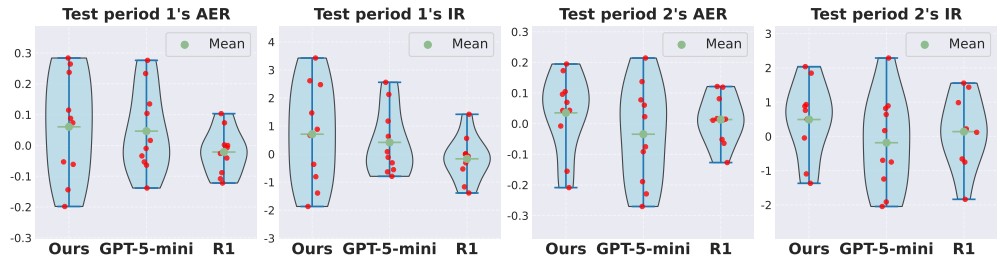

Figure 6: *Alpha* performance comparison with state-of-the-art LLMs on out-of-sample periods. Compared to two state-of-the-art LLMs, GPT-5-mini and DeepSeek-R1 (R1), our AlphaAgentEvo-4B demonstrates highly competitive performance, consistently achieving a higher average AER and average IR across test periods 1 and 2. This indicates that its evolution strategy exhibits favorable out-of-sample generalization and superiority under diverse market conditions. Meanwhile, its violin plots of test period 2 reveal a more evident top-heavy distribution, with a wider upper and narrower lower tail, suggesting that a greater share of its evolved *alphas* achieve positive predictive power.

## 4    RELATED WORK

Due to the vast number of available operators and features, the search space for *alphas* is astronomically large. A traditional category is Genetic Programming (GP), which generate candidates through random mutation and crossover (Lin et al., 2019; Schmidt & Lipson, 2010; Zhaofan et al., 2022; Patil, 2023), or introduce hierarchical mechanisms to identify reusable components for efficient search (Zhang et al., 2020). While GP can recycle partial structures from existing *alphas*, they are largely heuristic, fail to exploit feedback from failed candidates, and offer limited interpretability. Some Reinforcement Learning (RL)-based attempts (Yu et al., 2023; Shi et al., 2025a; Zhu & Zhu, 2025) further guide *alpha mining* with reward signals, but they still operate at the operator level and rely heavily on trial-and-error. In non-stationary markets, such incremental search is easily misled by spurious correlations and struggles to discover robust *alphas*.

Unlike conventional AI applications Zheng et al. (2026; 2024; 2023), Large language models (LLMs) provide a promising alternative by leveraging semantic reasoning and domain knowledge to construct more interpretable *alphas* (Wang et al., 2023; Haluptzok et al., 2023; Weng, 2023; Sumers et al., 2024; Shi et al., 2025b). Several recent studies (Luo et al., 2025; Wang et al., 2024) integrate LLMs into *alpha mining*, such as FAMA for dynamic factor combination (Li et al., 2024) and AlphaAgent (Tang et al., 2025) for a multi-agent architecture for decay-resistant *alphas*. However, these approaches remain essentially prompt-driven, lacking mechanisms for long-horizon planning, systematic reflection, and self-evolution. In parallel, RL-based LLM post-training (Wang et al., 2025; Jin et al., 2025; Chen et al., 2025) has made progress in mathematics, games, and tool use. Self-evolving LLM systems (ang Gao et al., 2025; Agrawal et al., 2025; Romera-Paredes et al., 2024; Chen et al., 2023) demonstrate great potential in solving complex problems through progressive refinement, but their application to alpha mining remains problematic, due to their vulnerability to market regime shifts or the inherent inability of text-based experience to fully encode desired evolutionary patterns. In this work, we bridge this gap with a self-evolving agentic RL paradigm.

## 5    CONCLUSION

In this work, we introduced AlphaAgentEvo, a novel agentic reinforcement learning paradigm for *alpha mining*. By reformulating alpha mining from a brute-force searching problem into a multi-turn

evolution-driven paradigm, our framework endows LLM-driven agents with stronger self-evolution capabilities, enabling them to dynamically extract structure from noisy and high-variance financial tool feedback through hierarchical reward signals. Extensive experiments on *AlphaEvo500* and *Alpha158* confirm that our method not only delivers consistently higher valid ratios and pass rates, but also generalizes effectively across market regimes and longer evolutionary trajectories, surpassing modern self-evolution approaches associated with state-of-the-art closed-source LLMs with only 4B parameters. These results highlight *self-evolving* ARL as a principled and generalizable paradigm for next-generation quantitative investment.

## 6 ACKNOWLEDGMENT

This work was supported in part by the National Natural Science Foundation of China (NSFC) under Grant 62276283, in part by the China Meteorological Administration's Science and Technology Project under Grant CMAJBGS202517, in part by Guangdong-Hong Kong-Macao Greater Bay Area Meteorological Technology Collaborative Research Project under Grant GHMA2024Z04, in part by Fundamental Research Funds for the Central Universities, Sun Yat-sen University under Grant 23hytd006, in part by Guangdong Provincial High-Level Young Talent Program under Grant RL2024-151-2-11, in part by the Key Development Project of the Artificial Intelligence Institute, Sun Yat-sen University, and in part by The Major Key Project of PCL (Grant No. PCL2025A17).

## REPRODUCIBILITY STATEMENT

In this submission, we have made extensive efforts to ensure the reproducibility of our work. Specifically, the dataset files used in this paper are enclosed in the supplementary materials, including training, validation, and test splits. The evaluation tool's parameters are described in Sec. 3.1 and the tool schema in Sec. F. Available data variables and representative functions are listed in Sec. G, Appendix. Training configurations are documented in Sec. D of the Appendix.

In addition, we provide the full source code (including training pipelines, evaluation scripts, and evaluation tools) as supplementary files. We will publicly release the source code associated with the used data once the paper is accepted to facilitate further research.

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

## A   DISCLOSURE OF LARGE LANGUAGE MODEL USAGE

In this paper, Large Language Models (LLMs) are only used for polishing paragraph content, checking and revising grammar, and writing visualization code for some experimental results during the paper writing process. All other parts of this paper were completed by human authors, in particular, the conception of research ideas, the creation of figures in the paper, the conduct of experiments, and the recording of experimental data.

## B   MULTI-FACTOR PERFORMANCE COMPARISON.

While some baseline approaches cannot be compared directly, to evaluate AlphaAgentEvo's performance with them, we evaluate AlphaAgentEvo based on the multi-factor strategy where a group of *alphas* are weighted to generate an meta *alpha*. We report the performance comparison results against three categories of baselines, including time series models (TS model), a non-LLM RL framework, and LLM-agent-based frameworks. For fairness, the top-10 mined alphas from each LLM-agent-based framework are evenly combined for backtesting.

Table 3: Multi-factor portfolio performance comparison from 2024-01 to 2025-11.

| Method | Category | Trainable | AER | IR | MDD |
|---|---|---|---|---|---|
| LightGBM (Ke et al., 2017) | TS model | ✓ | -0.009 | 1.192 | -0.195 |
| Stock-Mixer (Fan & Shen, 2024) | TS model | ✓ | 0.013 | 1.977 | -0.182 |
| AlphaQCM (Zhu & Zhu, 2025) | RL Framework | ✓ | 0.027 | 1.815 | -0.192 |
| AlphaAgent (Tang et al., 2025) | Multi-LLM-agent | ✗ | 0.064 | 2.046 | -0.196 |
| GPT-5-mini (OpenAI, 2025) | Single-LLM-agent | ✗ | -0.158 | 0.587 | -0.213 |
| ToolRL-4B (Qian et al., 2025) | Single-LLM-agent | ✓ | -0.027 | 1.532 | -0.215 |
| AlphaAgentEvo-4B (*ours*) | Single-LLM-agent | ✓ | **0.129** | **2.442** | **-0.176** |

## C   TRAINING ANALYSIS

To analyze the policy LLM's changes during the reinforcement learning process, we present the rewards in training and validation sets, average response length, and the output entropy during AlphaAgentEvo's training.

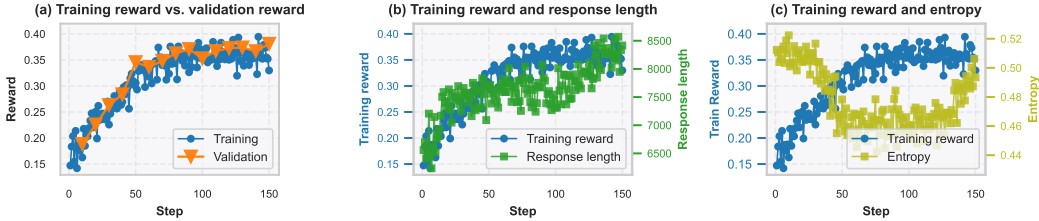

Figure 7: Training statistics of AlphaAgentEvo.

**Training reward vs. validation reward.**   Fig. 7(a), both training and validation reward curves rise steadily from ∼0.16 to ∼0.38–0.39 over 150 steps, with the validation reward closely tracking the training reward and maintaining a small generalization gap (visually < 0.02 throughout). The improvement is fastest in the first 50 steps and then saturates gradually, with minor jitter but no sign of overfitting: validation continues to trend upward in tandem with training.

**Response length.**   Fig. 7(b) shows a clear stepwise increase in average response length. Early on (0–50 steps), the model rapidly adapts to task requirements and learns to produce stable tool calls. Between 50–100 steps, growth slows as it improves tool-call quality and forms its own evolution strategy. After 100 steps, response length stabilizes at a relatively high level, reflecting the transition from basic adaptation to more complex reasoning for *alpha* mining.

**Output entropy.** As shown in Fig. 7(c), output entropy first decreases as the agent masters consistent reasoning, then remains stable, and finally rises again after $\sim 125$ steps. This rebound indicates renewed exploration, where the model diversifies its reasoning to generate richer *alphas*.

Taken together, these results verify that our training can converge stably with good generalization, and our method is capable of effective complex reasoning to mine richer and more effective *alphas*.

## D  TRAINING CONFIGURATIONS

We train Qwen3 (Yang et al., 2025) (1.7B and 4B) on $10 \times$ RTX4090 GPUs for 150 steps using the Verl framework (Sheng et al., 2024). The 4B model provides sufficient capability while maintaining a favorable performance–efficiency trade-off for large-scale *alpha mining*, with the 1.7B variant serving as a lighter comparison. Each batch samples 20 seed *alphas*, with 3 rollouts per seed, up to 3 turns per trajectory, and up to 4 tool calls per turn. The coefficient $\beta$ for KL loss is set to 0.001. The $80^{\text{th}}$ step checkpoint is used for testing. Reward caps are set as $C_{\text{tool}} = 1$, $C_{\text{cons}} = 0.2$, $C_{\text{expl}} = 0.3$, $C_{\text{perf}} = 0.5$, and $C_{\text{streak}} = 0.6$. The weighting coefficients are $\alpha_{\text{succ}} = 0.1$, $\alpha_{\text{fail}} = 0.2$, $\alpha_{\text{cons}} = 0.02$, $\alpha_{\text{exp}} = 0.02$, $\alpha_{\text{perf}} = 0.1$, and $\alpha_{\text{streak}} = 0.15$. We set the learning rate to $1 \times 10^{-6}$ with a warmup ratio of $0.1$, and use the Adam optimizer for training. The policy LLM is updated with a mini-batch size 20.

## E  CASE ANALYSIS

We take a sample from *Alpha158* as a case study (only first two turns) to showcase why our model outperforms other baselines in *alpha evolution*, as shown in Fig. 8.

In the first turn, two models demonstrate fundamentally different approaches to evaluating the seed *alpha*. The baseline model immediately suggests horizontal adjustments such as extending the lookback window or applying Z-score normalization, e.g., *"maybe using a longer window... or ZS-CORE..."*. This reflects a focus on technical re-scaling while leaving the underlying semantics of the *alpha* unchanged. By contrast, AlphaAgentEvo-4B begins with a critical examination of the seed *alpha*: *"I should check if the current* alpha *has any issues ... perhaps we can look at the cumulative effect of down days..."*. This reasoning explicitly questions the behavioral implication of the *alpha*—whether a higher value actually aligns with profitable trading logic. By identifying the misalignment between signal direction and trading outcome, AlphaAgentEvo-4B opens the path toward semantic reconstruction. This critical stance directly motivates subsequent refinements such as combining with RSI, normalizing by volatility, and embedding structural signals (e.g., ZigZag), which collectively transform a weak predictor into an *alpha* with strong positive alpha.

In the second turn, this divergence becomes even clearer. The baseline model reflects only on parameter choices from its previous trial: *"Looking at the previous attempts: the first new* alpha *SIGNIFI-CANT_DOWN_DAYS_90 ... But its IR is worse (-0.3529), so maybe the threshold is too strict or the window is too long."* The analysis remains confined to surface-level tuning, without questioning the semantic suitability of the *alpha* design. In contrast, AlphaAgentEvo-4B demonstrates a stronger capacity for reflective reasoning: *"RSI_DOWN_COUNT ... giving a high IR of 0.9417 ... the key here is that the previous factors improved by incorporating other indicators (RSI, volatility, swing points) with the down count..."* Rather than merely fine-tuning thresholds, AlphaAgentEvo-4B distills general patterns from past attempts and reasons about why certain designs succeed. Building on this reflection, it further proposes new factor constructions, such as adding industry neutrality or volatility adjustment, showing an ability to transform statistical feedback into structured insights that guide subsequent evolution.

Overall, the baseline model remains locked in mechanical parameter tweaks, while AlphaAgentEvo-4B demonstrates a dialectical process: integrating empirical outcomes with theoretical reasoning to refine the *alpha* semantics. This difference explains why our model converges to successful, high-performance *alphas* while the baseline model fails.

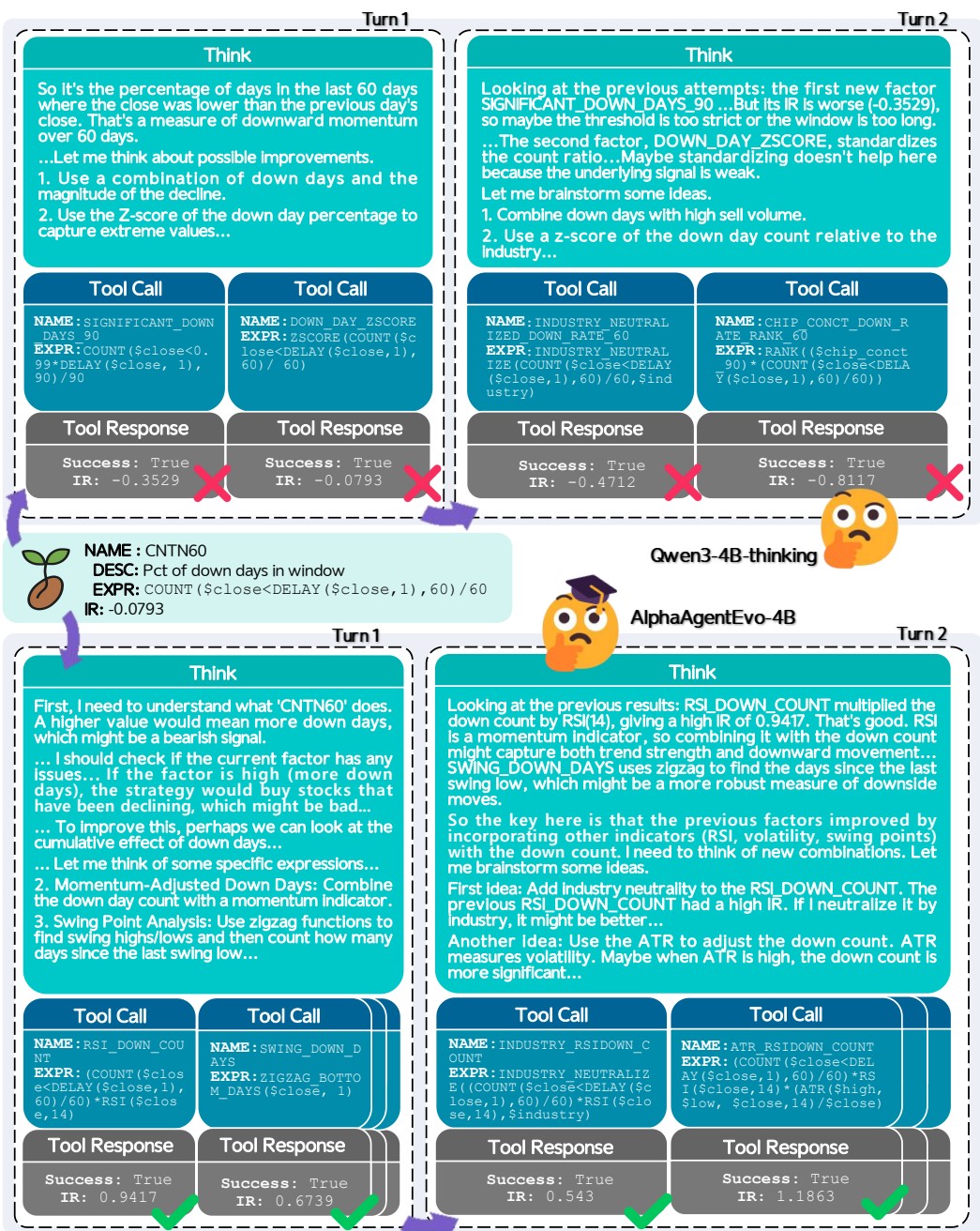

Figure 8: Case study: AlphaAgentEvo-4B vs. our base model Qwen3-4B-thinking.

## F  EVALUATION TOOL

As part of AlphaAgentEvo, we design a dedicated tool interface, `evaluate_factor`, to support the evaluation of *alphas* with backtesting. The simplified tool schema is shown in Listing 1, which illustrates the core arguments (`factor_name`, `factor_expr`, and `metric`). For clarity, some auxiliary parameters such as `time_range` and `market` are omitted here.

Listing 1: Schema of the `evaluate_factor` tool used in this paper.

```json
{
  "type": "function",
  "function": {
    "name": "evaluate_factor",
    "description": "A tool for evaluating factors with backtesting",
    "parameters": {
      "type": "object",
      "properties": {
        "factor_name": {
          "type": "string",
          "description": "The name of the factor"
        },
        "factor_expr": {
          "type": "string",
          "description": "The expression of the factor"
        },
        "metric": {
          "type": "string",
          "description": "The metric to evaluate (default: '
              Information_Ratio_with_cost')",
          "default": "Information_Ratio_with_cost"
        }
      },
      "required": ["factor_name", "factor_expr"]
    }
  }
}
```

## G  AVAILABLE DATA AND FUNCTIONS

Data variables used to construct *alphas* are shown in Table 5, all sourced from Tushare (Wang, 2024). A representative set of functions to operate these data variables are displayed in Table 4.

Table 4: Representative functions used in our *alpha* expressions.

| Function Name | Description |
|---|---|
| **A. Cross-Sectional Operations** | |
| RANK(var) | Cross-sectional percentile rank of a variable. |
| ZSCORE(var) | Standardizes a variable (z-score) cross-sectionally. |
| INDUSTRY_NEUTRALIZE(var, $industry) | Neutralizes the variable's exposure within industries. |
| **B. Time-Series / Rolling Window** | |
| TS_MEAN(var, p) | Rolling mean over the past 'p' periods. |
| TS_MAX(var, p), TS_MIN(var, p) | Rolling maximum and minimum. |
| TS_RANK(var, p) | Time-series percentile rank over a window. |
| TS_PCTCHANGE(var, p) | Percentage change over 'p' periods. |
| DELTA(var, p) | Difference from 'p' periods ago ($x_t - x_{t-p}$). |
| EMA(var, p), SMA(var, p) | Exponential and Simple Moving Average. |
| **C. Mathematical & Logical** | |
| LOG(var), POW(var, exp) | Natural logarithm and power. |
| DELAY(var, p) | Value of the variable 'p' periods ago (lag). |
| COUNT(cond, p) | Count of times a condition is true over 'p' periods. |
| A ? B : C | Ternary operator (if condition A then B, else C). |
| **D. Advanced & Technical** | |
| TS_CORR(var1, var2, p) | Rolling correlation between two variables. |
| REGBETA(var1, var2, p) | Rolling beta from regressing var1 on var2. |
| RSI(var, p) | Relative Strength Index. |
| MACD(var, p_short, p_long) | Moving Average Convergence Divergence. |

Table 5: Available data variables.

| Variable Name | Description |
|---|---|
| **Price & Market Data** | |
| $open | Opening price |
| $high | Highest price of the day |
| $low | Lowest price of the day |
| $close | Closing price |
| $volume | Trading volume (shares) |
| $amount | Trading amount (CNY) |
| $change | Price change vs. previous close |
| $return | Last day's return |
| **Chip-Distribution Data** | |
| $his_low | Historical low price since listing |
| $his_high | Historical high price since listing |
| $cost_5pct | Cost where 5% of chips lie below |
| $cost_15pct | Cost where 15% of chips lie below |
| $cost_50pct | Median cost of chips |
| $cost_85pct | Cost where 85% of chips lie below |
| $cost_95pct | Cost where 95% of chips lie below |
| $weight_avg | Average cost across all chips |
| $winner_rate | The chip win rate |
| $chip_conct_90 | Chip concentration within the densest 90% |
| $chip_conct_70 | Chip concentration within the densest 70% |
| **Order-Flow / Money-Flow Data** | |
| $buy_sm_vol, $sell_sm_vol | Small-lot buy/sell volume |
| $buy_sm_amount, $sell_sm_amount | Small-lot buy/sell turnover |
| $buy_md_vol, $sell_md_vol | Medium-lot buy/sell volume |
| $buy_md_amount, $sell_md_amount | Medium-lot buy/sell turnover |
| $buy_lg_vol, $sell_lg_vol | Large-lot buy/sell volume |
| $buy_lg_amount, $sell_lg_amount | Large-lot buy/sell turnover |
| $buy_elg_vol, $sell_elg_vol | Extra-large-lot buy/sell volume |
| $buy_elg_amount, $sell_elg_amount | Extra-large-lot buy/sell turnover |
| $net_mf_vol | Net inflow volume (buy - sell) |
| $net_mf_amount | Net inflow amount |
| **Benchmark & Industry** | |
| $bench_open, $bench_high, $bench_low, $bench_close | Benchmark index OHLC prices |
| $bench_preclose | Benchmark previous close |
| $bench_volume | Benchmark trading volume |
| $bench_amount | Benchmark trading amount |
| $bench_turn | Benchmark turnover ratio |
| $bench_return | Benchmark last day's return |
| $industry | Categorical industry label |

