# OpenReview forum: "AlphaAgentEvo: Evolution-Oriented Alpha Mining via Self-Evolving Agentic Reinforcement Learning"
_ICLR.cc/2026/Conference — ICLR 2026 Poster_

### Official Review · Reviewer_znBA · 2025-10-14

**Soundness:** 2
**Presentation:** 3
**Contribution:** 2
**Rating:** 2
**Confidence:** 5

**Summary:**

This paper introduces AlphaAgentEvo, a framework for quantitative alpha mining that uses Agentic Reinforcement Learning. This agent interacts with a financial backtesting tool over a multi-turn trajectory to progressively refine an initial "seed" alpha. The learning process is guided by a sophisticated, hierarchical reward function that balances tool usage correctness, structural consistency, exploration, performance improvement, and sustained progress. Experiments show that AlphaAgentEvo outperforms traditional Genetic Programming methods and state-of-the-art LLMs in generating alphas with higher success rates.

**Strengths:**

1. The hierarchical reward function is a key strength. Instead of relying on a single, sparse performance metric, it provides the agent with dense, multi-faceted feedback.
2. AlphaAgentEvo demonstrates superiority over a range of baselines, including GP and large reasoning models like DeepSeek-R1.

**Weaknesses:**

1. The core technical method is an adaptation of an existing RL algorithm (GRPO) to an agentic setting. The primary innovation lies in the problem formulation and reward engineering. This makes the contribution more of an applied nature than a fundamental advance.
2. True self-evolution might imply an agent that modifies its own learning algorithm or core reasoning architecture, whereas this work uses RL to improve performance on a task. The "evolution" observed is in the alpha's quality, not a fundamental evolution of the agent itself.
3. The experiment is not sufficient. It should be considered to compare models trained on this task. Currently, all the baselines are training-free.

**Questions:**

1. Why design this method just for alpha mining? Can it be performed on general tasks? What part of the work (except the reward) is specifically designed for alpha mining?
2. Why do you call your method "self"-evolving? It seems that the authors just use RL to train the LLM on the alpha mining task. There's no highlighted "self" part in the method. And the performance increase is evolving?

---

> ### Author Response · Authors · 2025-11-18
>
> **W1/Q1: The contribution lies in an applied nature more than a fundamental advance.**
>
> **Response**: Thank you for the suggestion. Existing Tool-RL approaches [1][2] focus on simple, static tools, which limit their effectiveness in dynamic financial markets characterized by rapid regime shifts and stochasticity. These methods struggle to provide actionable optimization signals for high-volatility environments.
>
> To address this, we reformulate tool use as a dynamic, evolution-driven process, moving away from the static paradigm. This innovative approach overcomes planning bottlenecks in existing alpha mining methods, enabling agents to **master complex financial tools**, **actively react to the underlying states behind tools**, and finally obtain **progressive improvements**. **To our best knowledge, AlphaAgentEvo is the first agentic RL approach for alpha mining**. Beyond quantitative improvements, AlphaAgentEvo exhibits **agent-level self-evolving behaviors** (please see below response or Section 3.3) that never appear in current appraoches. These properties highlight the core difference in how agents learn to interact with complex and dynamic financial tools that traditional methods cannot harness, rather than offering only an applied performance gain.
>
> ---
>
> **W2/Q2: The "evolution" observed is in the alpha's quality, not a fundamental evolution of the agent itself.**
>
> **Response**: The evolution in our framework is not merely reflected in the alpha outputs but in the agent's multi-turn behavior. **Our supplementary analysis (**Fig. 3, rebuttal revision**) provides further evidence**:
> - The mean IR gap between AlphaAgentEvo and a general tool-use RL baseline (ToolRL) continually widens over turns, which cannot be explained by alpha evolution alone. Rather, it suggests the **agent’s policy for generating alphas becomes progressively better**;
> - Higher and growing IR standard deviation as the number of turns increases, indicating an **adaptive expansion of its search horizon**;
> - Stable consistency relative to seeds, ensuring that improvements accumulate progressively rather than degenerating into uncontrolled search, showing that exploration is **not random trial-and-error but reveals a systematic strategy**;
>
> Such coordinated behavioral shifts are characteristic of **agent-level self-evolution**, rather than independent improvements in specific alpha candidates.
>
> ---
>
> **W3: It should be considered to compare models trained on this task.**
>
> **Response**: We appreciate the reviewer’s concern. **We additionally compare the following contemporary and strong baselines:**
> - A tool-use RL baseline ToolRL[1] (NeurIPS 2025)
> - An LLM-based evolutionary method GEPA [4] (configured with SoTA base LLMs);
> - A non-LLM-based RL method AlphaQCM [5] (ICML 2025);
> - Time series models for this task.
>
> Results are shown below:
>
> > **Table 1: Performance on AlphaEvo500 (24-25)**
> |Method|HS300-VR|HS300-P@3|HS300-P@5|CSI500-VR|CSI500-P@3|CSI500-P@5|
> |---|---:|---:|---:|---:|---:|---:|
> |GPT-5-mini|0.970|0.75|0.88|0.972|0.73|0.82|
> |DeepSeek-R1|0.872|0.68|0.71|0.886|0.71|0.86|
> |ToolRL-1.7B|0.864|0.74|0.78|0.851|0.66|0.74|
> |ToolRL-4B|0.954|0.75|0.81|0.961|0.73|0.76|
> |GEPA(GPT-5-mini)|**0.992**|0.87|0.90|0.971|*0.86*|*0.91*|
> |GEPA(DeepSeek-R1)|0.977|0.83|0.87|**0.978**|0.82|0.88|
> |AlphaAgentEvo-1.7B|0.940|0.77|0.90|0.923|0.76|0.78|
> |AlphaAgentEvo-4B|*0.979*|**0.97**|**0.97**|*0.977*|**0.93**|**0.95**|
>
> > **Table 2: Multi-factor Portfolio Performance Comparison (2024-01 to 2025-11)**
> |Method|Category|Trainable|AER|IR|MDD|
> |---|---|---|---:|---:|---:|
> |LightGBM (Ke et al.,2017)|Time-series Model|✔|-0.009|1.192|-0.195|
> |Stock-Mixer (Fan&Shen,2024)|Time-series Model|✔|0.013|1.977|-0.182|
> |AlphaQCM (Zhu et al.,2025)|Traditional RL|✔|0.027|1.815|-0.192|
> |AlphaAgent (Tang et al.,2025)|Multi-Agent|✘|0.064|2.046|-0.196|
> |GPT-5-mini (OpenAI,2025)|Single-Agent|✘|-0.158|0.587|-0.213|
> |ToolRL-4B (Qian et al.,2025)|Agentic RL|✔|-0.027|1.532|-0.215|
> |AlphaAgentEvo-4B (ours)|Agentic RL|✔|**0.129**|**2.442**|**-0.176**|
>
> [1] Qian, C., Acikgoz, E. C., He, Q., Wang, H., Chen, X., Hakkani-Tur, D., Tur, G., & Ji, H. (2025). ToolRL: Reward is all tool learning needs. Poster session presented at NeurIPS 2025.
>
> [2] Wang, Zihan, et al. "Ragen: Understanding self-evolution in LLM agents via multi-turn reinforcement learning." arXiv preprint arXiv:2504.20073 (2025).
>
> [3] Huan-ang Gao, H., Geng, J., Hua, W., Hu, M., Juan, X., Liu, H., ... & Wang, M. A survey of self-evolving agents: On path to artificial super intelligence, 2025. URL https://arxiv. org/abs/2507.21046.
>
> [4] Agrawal, L. A., Tan, S., Soylu, D., Ziems, N., Khare, R., Opsahl-Ong, K., ... & Khattab, O. (2025). Gepa: Reflective prompt evolution can outperform reinforcement learning. arXiv preprint arXiv:2507.19457.
>
> [5] Zhu, Z., & Zhu, K. (2025). AlphaQCM: Alpha discovery in finance with distributional reinforcement learning. In Proceedings of the 42nd International Conference on Machine Learning.

---

> ### Author Response · Authors · 2025-11-21
>
> **Dear Reviewer znBA**,
>
> Thank you again for your time and constructive comments.
>
> We are writing to kindly invite you to review our response, as we have conducted additional experiments and provided clarifications specifically addressing your concerns:
>
>
> - Technical Contribution (W1): We elaborated on how our dynamic tool-use reformulation solves specific planning bottlenecks in finance that standard Tool-RL methods cannot address.
>
>
> - Comparison with Trained Baselines (W3): You correctly pointed out the need to compare against trained models. In our revision (Table 1, Table 3, and Fig. 3), we have now added comparisons against ToolRL [1] (a general tool-use RL baseline), GEPA (An LLM-based evolutionary method), and AlphaQCM [5] (a trainable RL method for this task). Our method continues to demonstrate superior performance.
>
> - Self-Evolution Definition (W2): We clarified that "evolution" refers to the agent’s iterative improvement in its behavior over multi-turn interactions via active reflection or change in its internal state. Our supplementary experiments (Please see Section 3.3 or the above comment) reveal systematic behavioral improvements that emerge only at the agent-policy level. These results provide direct evidence that the evolution occurs within the agent itself rather than merely in the generated alphas.
>
>
> We hope these revisions and the new results address your concerns regarding the soundness and contribution of the work. We would greatly appreciate your feedback on these updates.
>
> Best regards,
> The Authors

---

### Official Review · Reviewer_5QHQ · 2025-10-25

**Soundness:** 2
**Presentation:** 3
**Contribution:** 2
**Rating:** 2
**Confidence:** 3

**Summary:**

The paper addresses the problem of _alpha mining_ — optimising financial factors (alphas) that demonstrate strong predictive power in markets. To this end, the authors propose _AlphaAgentEvo_, a reinforcement learning–based framework that learns an evolutionary process, essentially mapping an initial seed factor $f_{{seed}}$ to an optimal factor $f^*$, a process they term _alpha evolution_. Through experiments on real-world financial market data (CSI500), the authors report that AlphaAgentEvo outperforms existing baselines. Further analysis indicates that AlphaAgentEvo produces more diverse solutions and demonstrates stronger generalization to unseen data.

**Strengths:**

1. The paper is clearly written and accessible. Although I am not deeply familiar with the field of alpha mining, it effectively defines both the problem and the proposed solution methodology.

2. The application of RL + LLM evolution to the field of alpha-mining is interesting and novel.

3. The results are promising, particularly given that the evaluation was conducted on real-world, previously unseen financial data.

**Weaknesses:**

1. The paper argues that alpha evolution is a more effective approach for discovering \( f^* \) than static methods, as it “looks into past feedback.” It further claims that alternative “search–test–restart” strategies (which includes RL)—are “brittle.” However, RL methods, particularly off-policy methods, do in fact leverage prior experience to improve over time. Thus, the characterization of these methods as “naive” seems unconvincing. Further, the paper does not provide any evidence to support the claim of their “brittleness”.

    a. Further, in practice, AlphaAgentEvo uses 3 turns (which is quite low), so I’m curious as to how important the “evolution” is. For this, I suggest an ablation on the number of steps with t = 0 (predict f^* directly), t=1 (f_init -> f^*) and maybe longer horizons, to validate the effectiveness of “evolution”.


2. It is unclear how the reward function is “hierarchical.” From its definition in Eq. 5, it appears to simply combine multiple objectives. Moreover, the use of fractional forms for the first three rewards is not well justified—why not employ a simpler linear combination instead?

    a. (Minor) The reward function introduces several additional hyperparameters. An analysis of their robustness, along with a complete ablation study on all components of the reward function, would strengthen the practical validity of the proposed algorithm.

    b. What is the rationale for enforcing proximity to \( f_{\text{init}} \)? If the objective is solely to identify the optimal alpha \( f^* \), why constrain the search to remain close to the initial factor?

    c. How are the initial factors generated, and what is their quality or performance baseline?

3. (Datasets) The paper provides no details on how the dataset was constructed, which makes reproducibility and thorough analysis of the results difficult.

4. (Evaluation) Why do you use score > f_seed as the main evaluation and not score directly?

5. While I am not deeply familiar with the specifics of alpha mining, evaluating the model on training data (2023-01-01 to 2024-01-01) is a very poor choice for assessment. Moreover, dividing “seeds” into train, validation, and test sets is unconventional. The standard practice is to perform such splits on the data itself (i.e., train/validation/test data partitions). Therefore, I believe the results presented in Tables 1 and 2 (left), as well as Figure 3(a), should be interpreted with caution or potentially excluded from the evaluation.

6. (Unfair evaluation of baselines) I am concerned that the baseline methods may not have been allocated a comparable training budget. What is the training budget for your proposed method (e.g., 150 steps × k_t)? If so, what value of k_t was used? Was an equivalent budget applied to the baselines? From my understanding, it appears that the baselines have been given a significantly smaller budget (only three turns)

7. The paper lacks a baseline comparison with state-of-the-art LLM-based evolutionary methods, such as FunSearch, AlphaEvolve and EvoTune. Additionally, such work has not been discussed at all.

8. Figure 4: what is direction aware reward?

FunSearch: Romera-Paredes, B., Barekatain, M., Novikov, A. et al. Mathematical discoveries from program search with large language models. Nature

AlphaEvolve: Novikov, Alexander, et al. "AlphaEvolve: A coding agent for scientific and algorithmic discovery." arXiv preprint arXiv:2506.13131.

EvoTune: Šurina, A. et al. Algorithm discovery with large language models: Evolutionary search meets reinforcement learning. Second Conference on Language Modelling

**Questions:**

1. How important is the number of turns in the training of AlphaAgentEvo?
2. See Weakness 2 b,c, 4, 7 and 8.
3. How are the datasets constructed?
4. What is the training budget used for AlphaAgentEvo and the baselines?

---

> ### Author Response · Authors · 2025-11-18
>
> - W1: The effectiveness of evolution?
> - Response: We do not intend to characterize modern RL as brittle. Our use of “brittle search–backtest–restart cycle” targets existing multi-agent approaches that lack persistent evolution mechanisms, e.g., no long-horizon planning and no credit assignment across multi-step edits of alphas, rather than RL itself. AlphaAgentEvo, as an agentic RL instantiation, explicitly performs multi-turn reflection, planning, and tool calling..., forming a continuous evolution trajectory. The $T$-step ablation to isolate the effect of evolution is shown below:
>   |   Metric  | Seed Alphas | Gen 1  | Gen 2  | Gen 3  | Gen 4  | Gen 5  |
>   |--|--|--|--|--|--|--|
>   | **IR Mean** | 0.9004  | 1.6538 | 1.9193 | 2.0701 | 2.1800  | 2.2639  |
>   | **IR STD**  | 0.9676  | 0.8466 | 0.8241 | 0.8707 | 0.9051 | 0.9317  |
>
> ---
>
> - W2. Why not employ a simpler linear combination instead?
> - Response: The reward is “hierarchical” because its components are **ordered** and **structural**, not merely summed. The fractional term enforces basic feasibility (i.e., correct tool use, structural consistency, and diversity) and deems each the tool call as a cost to reduce unnecessary calls. This term may dominate the total reward before the agent can stably get higher-level performance rewards. The multiplicative term then activates only when feasible candidates are produced, preventing noisy performance metrics from dominating early search. A linear combination cannot impose this prerequisite structure, whereas the our hierarchical form explicitly encodes this structure.
>
> ---
>
> - W2b: Why enforce proximity to $(f_{\text{init}})$? Why not perform global search?
> - Response: Constraining the search around $(f_{\text{init}})$ is essential because unrestricted global exploration is statistically fragile and prone to overfitting in finite-sample backtesting. Proximity to $f_{\text{init}}$ acts as a regularizer that prevents the search from drifting into spurious, noise-driven formulas, and staying close to them preserves interpretability, which is critical for real-world risk management, monitoring, and portfolio governance. In practice, evolution around interpretable seeds yields far more stable and deployable alphas than unconstrained global search. A typical alpha generated from an unconstrained global search is: *"ABS(ABS(ABS(ABS(ABS(ABS(ABS(ABS(ABS(ABS(ABS(LOG(ABS(ABS(ABS(ABS(ABS(ABS($open))))))))))))))))))"*, which contains no meaningful or logically interpretable financial signal.
>
> ---
>
> - W2c/Q3: How the dataset was constructed?
> - Response: We curated 30 research directions by domain experts, and for each direction used GPT-4o to generate 20–30 candidate alphas. We then validated executability of alphas, removed near-duplicates expressions, and balanced performance metric signs (~1:1) to avoid selection bias. Upon acceptance, we will release the cleaned dataset splits, prompts, and execution/validation code to enable reproducibility.
>
> ---
>
> - W4: Why do you use $score > f_{seed}$ as the main evaluation and not score directly?
> - Response: Since this paper focuses on alpha mining through multi-turn evolution, we require a metric that directly reflects evolutionary improvement. Evaluating whether an evolved factor achieves a higher score than its seed provides a clear and comparable indicator of evolutionary success. In practice, a higher evolution success rate indicates that one can find more effective alphas given a limited computation budget and time. For alpha's performance comparison, please refer to Fig. 7 and Table 3 (Appendix in rebuttal revision), where our evolved alphas outperform global optimization (time-series models) and RL baselines.
>
> ---
>
> - W5: Why divide “seeds” into train, validation, and test sets?
> - Response: Given dynamic market data and a fixed operator set, the alpha mining task is to mine a set of low-correlated factors such that each factor attains strong IR on both visible and invisible time windows. In this setting, we intentionally start from a seed pool with expert-assigned directions (we explain the reason in W2b) and the agent evolves higher-performing descendants; splitting by seeds (train/val/test) therefore directly probes generalization across factor programs—i.e., whether the agent learns a seed-agnostic evolution strategy that consistently gains improvement given different starting points, rather than merely fitting one data slice. **We supplement a cross-market evaluation (24-25) in Table 1, making our conclusion more solid.** Finally, to ensure the agent does not collapse onto a common pattern that ignores seed semantics, we report diversity and low-correlation analyses (see Fig. 5), showing that evolved alphas remain diversified while improving performance, supporting our claim of generalization to different seeds.

---

> ### Author Response · Authors · 2025-11-18
>
> - W6/Q4: Training budget \& baseline parity.
>
> - Reponse: We clarify that our method uses a max of 3 interaction turns in training, with ($k_t$=4) tool calls per round (Sec. D, Appendix). All methods are allowed up to 4 tool calls per turn at test time, and we report 3-turn and 5-turn evolution results. Since there are no trainable LLM-based approaches for alpha mining so far, we compare non-LLM trainable baselines in Sec. C, Appendix. For AlphaQCM[2]/ToolRL[1], we match the training (i.e., identical training steps × max turns × $k_t$ using the same data) and inference budgets to ours, and select the best of 10 alphas in the training set. Results demonstrate that AlphaAgentEvo's alphas perform better with the multi-factor portfolio.
>
> ---
>
>
>
>
> **(@AC: For this weakness, please see the following comment, titled "Comparison to the latest LLM-based evolution method")**
> - W7: The paper lacks a baseline comparison with state-of-the-art LLM-based evolutionary methods.
>
> - Response: Taking FunSearch as an example, it relies on several key prerequisites explicitly stated in the original paper:
>   - **an efficient and deterministic evaluator**,
>   -  **a rich and low-noise scoring signal**, and
>   - **a fixed program skeleton with only a small isolated component to be evolved** (Nature 2024, Discussion).
> These conditions are crucial for enabling FunSearch’s million-scale evolutionary sampling and for ensuring that incremental mutations yield meaningful improvements.
>
>   However, **alpha mining fundamentally violates these prerequisites** and therefore cannot be directly compared with FunSearch.
>   - First, although backtesting on historical data is deterministic at the code level, **the goal of alpha mining is to produce alphas that remain robust under real-market randomness**, including execution slippage, liquidity impacts, market microstructure noise, and parameter instability. As a result, the evaluation signal is inherently **high-variance and sensitive to noise**, contradicting FunSearch’s requirement for a stable evaluator.
>   - Second, FunSearch typically requires **10⁶+ evaluations** to evolve a single function; in contrast, each alpha backtest is computationally expensive (milliseconds to seconds) and it is **completely infeasible** to perform millions of evaluations for a single candidate alpha.
>   - Third, unlike FunSearch’s small “critical function to evolve,” alpha formulas are **full symbolic expressions** without a stable skeleton, leading to an enormous discrete search space where almost all mutations produce invalid or meaningless financial signals.
>   - Finally, the objective in alpha mining is to generate **a set of low-correlated alphas**, not just optimize one program. This makes **iteration efficiency** and **diversity-aware exploration** essential—components that FunSearch does not support.
>
>   For these reasons, FunSearch and alpha mining operate under fundamentally different problem settings, and a direct comparison is not appropriate.
>
>   That said, we appreciate the reviewer’s suggestion. If deemed necessary, we can attempt an adapted variant of FunSearch for alpha mining and include preliminary results in the final version. However, such an adapted design may diverge substantially from the original FunSearch due to the intrinsic characteristics of alpha mining.
>
>
> ---
>
>
> - W8: Figure 4: what is direction aware reward?
> - Response: Thank you for pointing this out. The term “direction-aware reward” in Figure 4 is a typo. It should be a consistent reward. We have revised this in the rebuttal revision.
>
> ---
>
> - Q1: How important is the number of turns in the training of AlphaAgentEvo?
> - Response: We observe a clear performance improvement as the number of turns increases. This trend is expected because with more turns, the agent can better explore and assess the search space, progressively learning a more accurate estimation of state and action values to improve its final decision-making.
>
>
>
> [1] Qian, C., Acikgoz, E. C., He, Q., Wang, H., Chen, X., Hakkani-Tur, D., Tur, G., & Ji, H. (2025). ToolRL: Reward is all tool learning needs. Poster session presented at NeurIPS 2025. https://neurips.cc/virtual/2025/loc/san-diego/poster/116923
>
> [2] Zhu, Z., & Zhu, K. (2025). AlphaQCM: Alpha discovery in finance with distributional reinforcement learning. In Proceedings of the 42nd International Conference on Machine Learning.

---

> > ### Comment · Reviewer_5QHQ · 2025-11-18
> >
> > 1. Thanks for the ablation on the number of turns of evolution.
> >
> > 2. Still confused about the reward
> >
> >     a. Why is R inversely proportional to R_tool?
> >
> >     b.  I am not convinced that such a complex reward function is required. What happens with a simple linear reward function with tuned weights for each reward? Such an ablation is needed to establish the effectiveness of the proposed reward function
> >
> >     c. “multiplicative term then activates only when feasible candidates” – I am confused, where is feasibility being measured in the reward function?
> >
> > 3. Is the unconstrained alpha produced optimal in training? (high train / validation acc), It is essential to show quantitative results for this case as well
> >
> > 4. I’d like to request clarification on the paper’s actual objective. Given a dataset D and a feature matrix X, is the goal to find \arg\max f, or to develop an algorithm that efficiently hill-climbs starting from some f_{\text{seed}}? These are fundamentally different problems: the former corresponds to a standard reinforcement learning or supervised learning setup, while the latter aligns with a meta-learning formulation (e.g., MAML). From my understanding, the paper aims to address the latter; however, Equation (1) appears to describe the former, which seems inconsistent.
> >
> >     a. Given that the paper operates within a meta-learning framework, I still find it problematic to train and evaluate on the same dataset. If the model has access to \( f^* \), for instance, it could trivially predict it at \( t = 1 \), effectively overfitting to the training data. As the authors have stated in the rebuttal, this does not happen because they constrain the solution to be close to f_seed.
> >
> >     b. This leads me to my final question on this: why is f_seed -> f^* a bad solution if we are testing on the training dataset?
> >
> > 5. Thanks for the clarification on FunSearch. I agree that FunSearch is not a good baseline at test-time. My question is, instead of at test-time, can FunSearch be used at train-time to meta-learn an algorithm for evolution? For ex. via prompt tuning

---

> > > ### Author Response · Authors · 2025-11-21
> > > **Response to Q5**
> > >
> > > **Response to Q5**:
> > >
> > > We appreciate the insightful follow-up. Conceptually, we agree that one could consider using FunSearch at train time to meta-learn an evolution policy (e.g., via prompt tuning). In fact, AlphaAgentEvo already follows this algorithm-learning view: the agent learns how to evolve alphas across many seeds and market regimes, rather than memorizing specific formulas.
> > >
> > > However, using FunSearch as an outer-loop optimizer would still rely on multi-turn interation with the evaluation tool. In alpha evolution, each evaluation is not only **expensive** but also **high-variance** (due to sequential tool interaction / discrete semantics-sensitive mutations in alphas), contradicting FunSearch’s assumptions of a deterministic evaluator. This makes prompt-level search require millions of rollouts × N-time sampling for stablizing reward signals. Therefore, we view FunSearch-style prompt search as a complementary extension rather than a directly comparable baseline.

---

> > > > ### Comment · Reviewer_5QHQ · 2025-11-25
> > > >
> > > > > "on the alpha evolution problem setting"
> > > >
> > > > Thanks for the detailed response regarding the alpha evolution problem.
> > > >
> > > > That said, equation~(1) still does not fully specify the problem being solved:
> > > > (1) it makes no reference to $f_{\text{seed}}$, and
> > > > (2) it does not include any constraint or objective that encourages the final solution to remain close to $f_{\text{seed}}$, which I assume is the intended purpose of alpha evolution.
> > > >
> > > > > "on the applicability of Fun Search"
> > > >
> > > > FunSearch and other LLM based evolution methods have also been successfully applied to non-deterministic settings[1,2,3], typically using budgets far smaller than a million samples. Hence, it would therefore strengthen the paper to compare against these baselines and to demonstrate empirically---if true---that they are not sample-efficient in this setting.
> > > >
> > > > [1] Chen, A., Dohan, D., & So, D. (2023). Evoprompting: Language models for code-level neural architecture search. Advances in neural information processing systems, 36, 7787-7817.
> > > > [2] Agrawal, L. A., Tan, S., Soylu, D., Ziems, N., Khare, R., Opsahl-Ong, K., ... & Khattab, O. (2025). Gepa: Reflective prompt evolution can outperform reinforcement learning. arXiv preprint arXiv:2507.19457.
> > > > [3] Castro, P. S., Tomasev, N., Anand, A., Sharma, N., Mohanta, R., Dev, A., ... & Stachenfeld, K. L. (2025). Discovering symbolic cognitive models from human and animal behavior. bioRxiv, 2025-02.

---

> ### Author Response · Authors · 2025-11-21
> **Response to Q2/Q3/Q4**
>
> **Response to Q2 (a–c)**:
>
> **Our reward function is designed to prevent brute-force search through frequent tool calls, while encouraging meaningful and efficient alpha evolution.** The tool-use term $(R_{cons} + R_{expl}) / R_{tool}$ treats each backtest call as a **cost**, so rewards increase only when the agent produces executable and semantically valid alphas with minimal redundant evaluations.
>
> Regarding the reward structure, simple linear combinations were tested but led to **reward hacking**: when the performance term is heavily weighted (e.g., large $R_{perf}$), the agent may ignore exploration and consistency and repeatedly exploit a few high-performance alphas. Such an agent may lose alpha diversity given different seeds (just remember some "good alphas") and may fail to generalize to different periods/seeds/markets. Exploration without performance is useless, while performance without stability or diversity can be risk-taking. The multiplicative formulation enforces joint satisfaction after RL training, allowing the agent to progressively improve these aspects and finally simultaneously master them, avoiding collapse into a small set of noisy super-alphas. We will include linear rewards' ablations in the later version.
>
>
> **Response to Q3/Q4**:
>
> We apologize for any confusion earlier. To clarify: unconstrained alpha mining (i.e., directly searching for the best formula in the full symbolic space) behaves similarly to genetic programming (GP) and is prone to overfitting during backtesting. As illustrated in the table below, both raw data optimization (using LGBM with early stopping) and unconstrained alpha formula search (GP) yield very high, or seemingly competitive, training Information Coefficient (IC) and Information Coefficient Information Ratio (ICIR)—both metrics where higher values are better. However, they perform poorly on validation and test sets, with GP even resulting in negative values. In contrast, our seed-constrained evolution achieves a lower training IC but preserves and even enhances IC and ICIR on unseen data.
>
> | Method | Description | Training Set | Validation Set | Test Set |
> |--------|-------------|--------------|----------------|----------|
> | | | **IC** / **ICIR** | **IC** / **ICIR** | **IC** / **ICIR** |
> | **LGBM (raw data)** | Global optimization | 0.2031 / 1.5296 | 0.0140 / 0.1073 | 0.0087 / 0.0661 |
> | **GP (mean of 10 alphas)** | Global optimization in formulaic space | 0.0355 / 0.5031 | -0.0208 / -0.3082 | 0.0010 / 0.0109 |
> | **Ours (mean of 10 alphas)** | Seed-constrained optimization | 0.0177 / 0.2152 | 0.0120 / 0.1293 | 0.0158 / 0.1502 |
>
>
> This highlights that alpha mining is not a supervised learning problem of directly solving $\arg\max f$; without structural constraints, optimal solutions often exploit noise. **Previous studies [1][2] have shown that fewer than 3% of discovered *alphas* remain effective out-of-sample.**
>
> Our motivation is grounded in a simple financial principle: no single factor can fully explain market returns (here “factor” means a single standalone driver). Asset prices are influenced by various factors—macro cycles, liquidity flows, investor sentiment, and temporary supply-demand imbalances. Since these factors change over time, a factor that works today may fail tomorrow, and a purely “optimal” factor often just exploits past noise.
>
> Thus, effective alpha mining focuses on continuously improving interpretable building blocks that can be monitored, combined, and managed in a multi-factor portfolio. By evolving new alphas from meaningful seeds rather than randomly searching for mathematical optima, we maintain economic intuition, avoid noise-driven artifacts, and develop factors that are profitable, understandable, and deployable in real investment systems.
>
> In summary, our goal is not to find the “best” $f$ on a dataset, but to **learn a meta-policy that progressively and interpretably (adjacently) evolves expert-assigned seed alphas into stronger neighborhoods**. **We have updated Section 2.1 including the overall objective to clarify this distinction.**

---

> ### Author Response · Authors · 2025-11-28
> **Comparison to the latest LLM-based evolution method**
>
> We compare our method to the latest and strongest baseline, GEPA, which is configured with GPT-4-mini and DeepSeek-R1 as its adapter and reflection LLM. The scoring function follows Eq. (6), and the evolutionary object is the prompt fragment referring to the evolution strategy. This comparison employs identical training/validation sets and evaluation budgets (i.e., max metric calls) to ensure a fair assessment. We have added these results and discussion on other LLM based evolution methods to our rebuttal revision.
>
> Results are shown below:
>
> > **Table 1: Performance on AlphaEvo500 (24-25)**
> | Method | HS300-VR | HS300-Pass@3 | HS300-Pass@5 | CSI500-VR | CSI500-Pass@3 | CSI500-Pass@5 |
> |--------|----------|--------------|--------------|-----------|---------------|---------------|
> | GPT-5-mini | 0.970 | 0.75 | 0.88 | 0.972 | 0.73 | 0.82 |
> | DeepSeek-R1 | 0.872 | 0.68 | 0.71 | 0.886 | 0.71 | 0.86 |
> | ToolRL-1.7B | 0.864 | 0.74 | 0.78 | 0.851 | 0.66 | 0.74 |
> | ToolRL-4B | 0.954 | 0.75 | 0.81 | 0.961 | 0.73 | 0.76 |
> | GEPA (GPT-5-mini) | **0.992** | 0.87 | 0.90 | 0.971 | *0.86* | *0.91* |
> | GEPA (DeepSeek-R1) | 0.977 | 0.83 | 0.87 | **0.978** | 0.82 | 0.88 |
> | AlphaAgentEvo-1.7B | 0.940 | 0.77 | 0.90 | 0.923 | 0.76 | 0.78 |
> | AlphaAgentEvo-4B | *0.979* | **0.97** | **0.97** | *0.977* | **0.93** | **0.95** |
>
>
> > **Table 2: Performance on Alpha158 (24-25)**
> | Method             | VR        | Pass@3    | Pass@5    |
> | ------------------ | --------- | --------- | --------- |
> | DeepSeek-R1        | 0.874     | 0.872     | *0.943* |
> | GPT-5-mini         | 0.975     | 0.828     | 0.903     |
> | GEPA (DeepSeek-R1) | 0.946     | 0.856     | 0.887     |
> | GEPA (GPT-5-mini)  | **0.989** | 0.836     | 0.881     |
> | AlphaAgentEvo-1.7B | 0.917     | *0.909* | 0.926     |
> | AlphaAgentEvo-4B   | *0.982* | **0.963** | **0.994** |
>
>
>
> > Original prompt fragment:
> ```
> Adhere to the following steps to achieve this objective:
> - Carefully analyze the existing factor expressions and their results, paying special attention to how each component or term within the expressions contributes to the overall performance. Use this detailed analysis to guide the evolution and design of the next generation of factors, aiming to enhance the target metric.
> - A metric value equal to nan or 0 means an error occurs when backtesting the factor or no signal is triggered. You need to double-check the expression.
> - Use the available variables and functions to craft new factors with the highest possible value for the target metric.
> - You are encouraged to design factors with complex and sophisticated logic, for example, incorporating up‑down price structure over long-period windows.
> ```
>
> ---
>
> > GEPA-evolved prompt fragment:
> ```
> ...
> - Signal design guidance:
>   - Favor robust, well-covered signals. Avoid conditions that return nan for most dates or symbols.
>   - Normalize or rank signals (ZSCORE, TS_ZSCORE, RANK) to stabilize magnitude and cross-section comparability.
>   - Use regime detection and structure:
>   ...
> - Syntax care:
>   - Use available function names exactly as supported (e.g., TS_ZSCORE vs ZSCORE where appropriate).
>   - Ensure parentheses and operator precedence are correct.
>   - Avoid divisions that can lead to instability (e.g., TS_STD(...) near zero), and prefer normalized measures or add gating/confirmation to avoid pathological cases.
> - Iteration strategy (use feedback if provided):
>   - Start by lightly evolving the initial factor (adjust lookbacks, add normalization/smoothing).
>   - Propose multiple diverse variants (3–6) that explore:
>     - Normalization and ranking
>     - Regime detection tweaks (e.g., changing window lengths, using DELTA on correlation/vol)
>     - Volume/momentum confirmations and gating thresholds
> ...
> ```
>
> We acknowledge that GEPA achieves improvements over its original prompt, but its evolution may inherit some limitations for alpha mining. First, as seen in Table 2 above, such prompt-based evolution **struggles under distributional shifts**, whereas our agentic approach continues to improve. Moreover, many **desired evolutionary patterns** (such as `synonymous substitution`, i.e., prefer a concise combination of operators to capture market anomalies) **cannot be exhaustively specified through natural-language prompts** but can be easilier learned through self-play, further highlighting limitations of GEPA-like methods.

---

> ### Author Response · Authors · 2025-11-28
> **On the alpha evolution problem setting**
>
> We thank the reviewer for pointing out that the original formulation did not clearly specify the role of the seed factor or the locality constraint. In the revised version:
>
> - Eq. (1) makes clear that alpha evolution learns a policy that performs **efficient local neighborhood search** around each seed by enforcing $\mathrm{sim}(f,f_{\text{seed}})\le\delta$, rather than unconstrained global optimization.
> - The objective maximizes the expected quality of the best evolved candidate generated by the policy over both in-distribution and out-of-distribution market regimes, **emphasizing that we train a general evolution policy (meta-policy), not a standalone factor.**
>
> Together, these clarify that the goal of alpha evolution is to learn an agent capable of efficient, seed-conditioned local exploration, producing stronger, reliable, and interpretable alphas—rather than solving an unconstrained optimization problem over the entire search space.

---

### Official Review · Reviewer_68ro · 2025-10-31

**Soundness:** 3
**Presentation:** 4
**Contribution:** 3
**Rating:** 6
**Confidence:** 3

**Summary:**

The paper proposes AlphaAgentEvo, a self-evolving Agentic RL (ARL) framework that turns alpha mining from a brittle search–backtest–restart loop into a multi-turn evolution process guided by a hierarchical reward (tool-use validity, direction-aware consistency to the seed, exploration diversity, performance, and improvement streak). The policy LLM plans, proposes multiple offspring factors per turn, queries a backtesting tool, reflects on outcomes, and updates via a GRPO-style objective adapted to multi-turn, tool-in-the-loop trajectories. Experiments on AlphaEvo500 (new) and Alpha158 across bearish/bullish periods report higher valid ratios and pass rates than GP, multi-agent, and strong LLM baselines, with 1.7B–4B models outperforming larger closed models on several metrics.

**Strengths:**

1.	Well-motivated paradigm: Precise problem framing of alpha evolution and limitations of GP / prompt-only methods; clear move from single-shot search to multi-turn evolution with reflection.
2.	Methodical ARL design: A GRPO-style objective extended to multi-turn trajectories and masked tool tokens; multiple offspring per turn with group-normalized advantages.
3.	Hierarchical reward that encodes domain priors: direction-aware similarity, exploration via AST-based structural diversity, performance and streak bonuses, with caps to prevent domination.
4.	Strong empirical evidence across two libraries and two regimes, with consistent gains in VR and pass@T; AlphaAgentEvo-4B reaches pass@5 of 0.76/0.94 on AlphaEvo500 and 0.725/0.994 on Alpha158.

**Weaknesses:**

1.	Many hand-set caps/weights and thresholds (e.g., similarity thresholds, reward caps C*, α*)—no sensitivity study is shown; robustness to these knobs is unclear.
2.	Heavy use of pass@T and VR; fewer statistics on effect sizes with uncertainty (CI/std) and transaction-cost / turnover analyses. Some violin plots are helpful but more rigorous statistical testing is desirable.
3.	GP incompatibility with AlphaEvo500 and varying offspring budgets complicate strict apples-to-apples comparisons; stronger head-to-head with exactly matched toolchains would isolate the contribution of ARL vs. retrieval/evaluation differences.
4.	The method is framed as broadly agentic, but all experiments remain alpha-formula evolution; no results on portfolio construction, risk models, or cross-market transfer.

**Questions:**

1.	Report compute per trajectory, backtest latency, and throughput relative to baselines at equal offspring budgets.
2.	When do direction-aware and exploration rewards conflict? Please show concrete failure trajectories and mitigations.

---

> ### Author Response · Authors · 2025-11-18
>
> **W2: Heavy use of pass@T and VR; fewer statistics on effect sizes with uncertainty (CI/std) and transaction-cost / turnover analyses.**
>
> **Response**: In the rebuttal revision, we supplement per-turn evolution statistics in Section 3.3, where the mean IR difference between ours and a strong tool-based RL baseline is enlarged over time. Also, AlphaAgentEvo demonstrate an growing standard deviation as the turn number increases, indicating its aggressive exploration in the search space.
>
> We additionally report effect sizes and uncertainty estimates for the IR improvement (ΔIR) of AlphaAgentEvo over ToolRL. Starting from the second turn, the effect size (Cohen’s *d*) increases steadily from **0.34 → 0.50**, showing that the performance advantage becomes progressively stronger as the agent evolves. Moreover, the **95% confidence intervals** of ΔIR remain **strictly positive** for all turns:
>
> > **Table 1: 95% CI Across Evolution Turns**
> | Turn | 95% CI (ΔIR) |
> |------|---------------------|
> | 1    | [0.12, 0.22] |
> | 2    | [0.24, 0.34] |
> | 3    | [0.29, 0.39] |
> | 4    | [0.34, 0.45] |
> | 5    | [0.39, 0.51] |
>
> These results demonstrate that the gains are statistically significant rather than arising from variance in factor evaluation. The **increasing effect sizes** and **expanding positive CIs** provides strong quantitative evidence that AlphaAgentEvo’s improvement reflects an increasingly effective **agent-level evolution strategy**, not merely isolated alpha-level modifications.
>
>
> Further, we conduct a portfolio performance comparison in Sec. C, Appendix, where different categories of approaches are evaluated in terms of annualized excess return, information ratio, and max drawdown. In this comparison, the top 10% of stocks are selected for buying long for each period, **their turnover and transaction costs stay very close**. If further statistical comparisons are desired, we would be happy to include them upon the reviewer’s guidance.
>
> > **Table 2: Multi-factor Portfolio Performance Comparison (2024-01 to 2025-11)**
> |Method|Category|Trainable|AER|IR|MDD|
> |---|---|---|---:|---:|---:|
> |LightGBM (Ke et al.,2017)|Time-series Model|✔|-0.009|1.192|-0.195|
> |Stock-Mixer (Fan&Shen,2024)|Time-series Model|✔|0.013|1.977|-0.182|
> |AlphaQCM (Zhu et al.,2025)|Traditional RL|✔|0.027|1.815|-0.192|
> |AlphaAgent (Tang et al.,2025)|Multi-Agent|✘|0.064|2.046|-0.196|
> |GPT-5-mini (OpenAI,2025)|Single-Agent|✘|-0.158|0.587|-0.213|
> |ToolRL-4B (Qian et al.,2025)|Agentic RL|✔|-0.027|1.532|-0.215|
> |AlphaAgentEvo-4B (ours)|Agentic RL|✔|**0.129**|**2.442**|**-0.176**|
>
>
> ---
>
> **W3/W4: Head-to-head comaparison.**
>
> **Response**: We thank the reviewer for highlighting the importance of strict head-to-head comparisons. We additionally evaluated GP under the same offspring budget (4 offspring per generation) used by all other baselines. Importantly, all methods in our study share the same evaluation toolchain, including an identical backtesting codebase, data splits, and transaction-cost settings. Also, as mentioned above, we conduct a head-to-head portfolio performance comparison where all approaches share a unified setting. We will release our source code for reproduction after the paper is accepted.
>
> ---
>
> **W4: Results on risk models, or cross-market transfer.**
>
> **Response**: We evaluate cross-market transferability in Table 1 left, where evolved alphas exhibit consistent improvements when transferred from the CSI500 to the HS300 market, suggesting that the agentic mechanism is not confined to a single market or environment. Extending the framework to risk modeling is indeed a necessary step towards real-world application. We view these components as important follow-up directions, and we plan to incorporate exposure-aware objectives, factor-neutral constraints (now `INDUSTRY_NEUTRALIZE` serves as an operator), and risk-adjusted optimization into the next stage of our work.
>
> ---
>
> **W1: Many hand-set caps/weights and thresholds and no sensitivity study is shown.**
>
> **Response**:  We thank the reviewer for the insightful comment. The hand-set caps and thresholds are not intended to tune performance, but rather to **prevent reward-hacking behavior** such as reward exploitation and to ensure the agent first acquires fundamental abilities before shifting to higher-level competencies such as planning, strategy formation, and long-horizon improvement. In practice, **these caps act as protective bounds and have a very limited impact on the agent’s evolutionary behavior**.
>
> Regarding sensitivity studies, a full sensitivity study is not feasible within the rebuttal period, as a single AlphaAgentEvo training run requires approximately 2 GPU-days. We will include a robustness analysis in the final version.

---

> ### Author Response · Authors · 2025-12-03
>
> **Q1: Report compute per trajectory, backtest latency, and throughput relative to baselines at equal offspring budgets.**
>
> **Response:** We report the computational metrics per optimization turn. Under a fixed offspring budget (the maximum tool calls per turn across all models), differences in total trajectory compute relative to baselines are determined solely by the LLM decoding time (Observing/Prefilling, Thinking, and Tool Calling). Below shows our pipeline latency per turn:
>
> | Metric | Time | Contribution to Total Latency |
> | :--- | :--- | :--- |
> | **LLM Decoding Time** | **33.42s** | **52.5%** (Model Dependent) |
> | **Tool Execution Time** | 30.25s | 47.5% (Fixed Environment) |
> | **Total Turn Latency** | **63.67s** | 100% |
>
>
> The backtest engine acts as the primary constraint on system throughput.
> *   **Backtest Latency:** **8.79s** per tool call (average).
> *   **System Throughput:** **0.11 calls/s**.
>
>
> ---
>
>
> **Q2: When do consistency rewards and exploration rewards conflict?**
>
> **Response:** The conflict between *Consistency Rewards* (penalizing low similarity to the seed) and *Exploration Rewards* (penalizing high similarity to known factors) creates a "narrow search corridor." When combined with tool noise, the agent is forced to abandon valid, simple paths within this corridor and resort to overfitting.
>
> We illustrate this with the `MIN20` (Failure) vs. `MIN10` (Success) trajectories. Both seeds started with "Low/Reversal" logic.
>
> - In the `MIN20` case, the conflict was triggered by **misleading tool responses**:
>   - **The valid attempt:** The agent generated alphas like `MIN60` with minimal editions. This perfectly satisfied the *Consistency Reward* (structural similarity) and the *Exploration Reward* (distinct parameter). However, the high-variance tool returned all degraded scores, disencouraging the agent from discovery of many potential editions implied by these alphas.
>
>   -  **The trap:** Since the simple exploration path was rejected by the tool, and the **Exploration Reward** prevented reverting to the seed, the agent was forced into **complexity overfitting**. It added more layers of `ATR` and `SLOPE` to satisfy the requirement for exploration and consistency while desperately trying to improve performance, leading to failure.
>
> - In contrast, the `MIN10` case benefited from **initial randomness**, where its simple variation (`RECOVERY_30`) bypassed the noise barrier, allowing it to optimize linearly without adding complexity.
>
> To mitigate this, a **consistency decay mechanism** can be introduced to progressively relax similarity constraints, potentially allowing the agent to escape these noise-induced local traps.

---

### Official Review · Reviewer_MHr3 · 2025-11-01

**Soundness:** 3
**Presentation:** 3
**Contribution:** 3
**Rating:** 6
**Confidence:** 2

**Summary:**

The paper introduces AlphaAgentEvo, a self-evolving Agentic Reinforcement Learning (ARL) framework designed for alpha mining—the process of discovering quantitative trading signals (“alphas”) that predict stock returns. This is the first work to propose a self-evolving agentic reinforcement learning framework for training LLMs for quantitative alpha mining. The performance of training Qwen3-1.7B with the proposed method is significantly better than prompting stronger models including GPT5, and GPT3.5 with the AlphaAgent framework.

**Strengths:**

1. This is the first work to propose a self-evolving agentic reinforcement learning framework for training LLMs for quantitative alpha mining.

2. The training algorithm is modified on top of GRPO and the rewards are carefully designed with ablation study showing their effectiveness.

3. The performance of training Qwen3-1.7B with the proposed method is significantly better than prompting stronger models including GPT5, and GPT3.5 with the AlphaAgent framework.

**Weaknesses:**

1. Please define alpha mining in the abstract/introduction.

2. As I'm not familiar with alpha mining, not sure if there are any related training-based LLM baseline for alpha mining (seems to be none). If there is any, should be used as a baseline as well.

3. There is only one training set and two test set used for experiments. More evaluation benchmarks would make the results more convincing.

**Questions:**

See weaknesses.

---

> ### Author Response · Authors · 2025-11-18
>
> **W1: Please define alpha mining in the abstract/introduction.**
>
> **Response**: Thank you for the suggestion. We have now added a clear definition of alpha mining in both the abstract and introduction. Specifically, we define it as: **"Alpha mining refers to uncovering quantitative signals that generate excess returns beyond the market."**
>
> ---
>
> **W2: Any training-based LLM baseline?**
>
> **Response**: To our best knowledge, we are the first tool-based/agentic RL method specialized for this task. In Table 1 (Rebuttal revision), we further compare a general tool-augmented RL baseline, ToolRL[1] and an LLM-driven reflective prompt evolution framework, GEPA[2]. Results are shown below:
>
> > **Table 1: Performance on AlphaEvo500 (24-25)**
> | Method | HS300-VR | HS300-Pass@3 | HS300-Pass@5 | CSI500-VR | CSI500-Pass@3 | CSI500-Pass@5 |
> |--------|----------|--------------|--------------|-----------|---------------|---------------|
> | GPT-5-mini | 0.970 | 0.75 | 0.88 | 0.972 | 0.73 | 0.82 |
> | DeepSeek-R1 | 0.872 | 0.68 | 0.71 | 0.886 | 0.71 | 0.86 |
> | ToolRL-1.7B | 0.864 | 0.74 | 0.78 | 0.851 | 0.66 | 0.74 |
> | ToolRL-4B | 0.954 | 0.75 | 0.81 | 0.961 | 0.73 | 0.76 |
> | GEPA (GPT-5-mini) | **0.992** | 0.87 | 0.90 | 0.971 | *0.86* | *0.91* |
> | GEPA (DeepSeek-R1) | 0.977 | 0.83 | 0.87 | **0.978** | 0.82 | 0.88 |
> | AlphaAgentEvo-1.7B | 0.940 | 0.77 | 0.90 | 0.923 | 0.76 | 0.78 |
> | AlphaAgentEvo-4B | *0.979* | **0.97** | **0.97** | *0.977* | **0.93** | **0.95** |
>
>
> Besides, we take a non-LLM-based RL approach, AlphaQCM [3], and trainable time-series forecasting models as supplementary baselines, comparing their overall portfolio performance (as they are not evolution-based). These results are shown as follows (also in Section B, Appendix):
>
> > **Table 2: Multi-factor Portfolio Performance Comparison (2024-01 to 2025-11)**
> |Method|Category|Trainable|AER|IR|MDD|
> |---|---|---|---:|---:|---:|
> |LightGBM (Ke et al.,2017)|Time-series Model|✔|-0.009|1.192|-0.195|
> |Stock-Mixer (Fan&Shen,2024)|Time-series Model|✔|0.013|1.977|-0.182|
> |AlphaQCM (Zhu et al.,2025)|Traditional RL|✔|0.027|1.815|-0.192|
> |AlphaAgent (Tang et al.,2025)|Multi-Agent|✘|0.064|2.046|-0.196|
> |GPT-5-mini (OpenAI,2025)|Single-Agent|✘|-0.158|0.587|-0.213|
> |ToolRL-4B (Qian et al.,2025)|Agentic RL|✔|-0.027|1.532|-0.215|
> |AlphaAgentEvo-4B (ours)|Agentic RL|✔|**0.129**|**2.442**|**-0.176**|
>
>
> In Section 3.3, we also conduct evolution analysis against ToolRL, demonstrating agent’s alpha generation policy becomes progressively better, for example, by adaptive expansion of its search horizon.
>
> ---
>
> **W3: More evaluation benchmarks?**
>
> **Response**: **We have added HS300 stock universe's results in Table 1 left (as shown above)**. For more alpha zoo's results, we are currently extending our experiment results to the Alpha360 benchmark [4]. Due to the large number of alphas involved and long evolution trajectories, these experiments require additional computation time. We will include the complete Alpha360 results in the final version of our submission.
>
>
> [1] Qian, C., Acikgoz, E. C., He, Q., Wang, H., Chen, X., Hakkani-Tur, D., Tur, G., & Ji, H. (2025). ToolRL: Reward is all tool learning needs. Poster session presented at NeurIPS 2025. https://neurips.cc/virtual/2025/loc/san-diego/poster/116923
>
> [2] Agrawal, L. A., Tan, S., Soylu, D., Ziems, N., Khare, R., Opsahl-Ong, K., ... & Khattab, O. (2025). Gepa: Reflective prompt evolution can outperform reinforcement learning. arXiv preprint arXiv:2507.19457.
>
> [3] Zhu, Z., & Zhu, K. (2025). AlphaQCM: Alpha discovery in finance with distributional reinforcement learning. In Proceedings of the 42nd International Conference on Machine Learning.
>
> [4] Microsoft. Qlib: An AI-oriented quantitative investment platform. GitHub. https://github.com/microsoft/qlib

---

### Comment · Area_Chair_up1A · 2025-11-25

Dear Reviewer, thank you for reviewing for ICLR. Since the discussion deadline is coming soon, could you please take a look at the author's rebuttal, respond to their comments, and update your rating as well? Thanks!

---

### Author Response · Authors · 2025-11-28
**Response to All Reviewers**

Dear Reviewers:

We sincerely thank all reviewers for their time, careful reading, and constructive feedback. Your comments significantly help us clarify the problem setting, strengthen the experimental design of the paper.

During the rebuttal stage, we made several substantial improvements.

First, we refined the formulation of the alpha evolution problem by explicitly modeling it as **policy-based local neighborhood search** around initial seeds, introducing the policy-induced candidate set and an explicit similarity constraint. Second, we expanded our evaluation by incorporating **stronger and more diverse baselines**, including:
- A post-training RL baseline: ToolRL (NeurIPS 2025).
- An LLM-based evolution method: GEPA (2025), configured with state-of-the-art LLMs.
- Global optimization approaches directly trained on this task, including AlphaQCM (ICML 2025), StockMixer (AAAI 2024).

Next, we conducted an **evolution analysis**. The results demonstrate that AlphaAgentEvo achieves accelerating IR gains and adaptive exploration, suggesting an evolving agent that improves itself with experience. This provides empirical evidence of agent-level self-evolution, distinguishing our approach from simple alpha-level optimization.

Most importantly, we emphasize the core contribution: AlphaAgentEvo reformulates financial tool use from a static, single-shot paradigm into a dynamic, evolution-driven process. Existing Tool-RL approaches rely on simple, static tools and struggle in financial markets characterized by stochasticity, rapid regime shifts, and weak or noisy optimization signals. By contrast, our formulation enables an agent to **master complex financial tools through iterative evolution, overcoming planning bottlenecks** that limit traditional alpha mining pipelines. **To our best knowledge, AlphaAgentEvo is the first agentic RL framework for alpha mining**. We view this reformulation as a step toward deploying agentic RL in more complex tool-centric tasks.

We thank all reviewers again for their insightful suggestions, which have meaningfully improved the clarity and strength of our work.

---

### Meta-Review · Area_Chair_7cwm · 2025-12-25

**Summary:**

This paper proposes AlphaAgentEvo, an agentic reinforcement learning framework that reformulates alpha mining as a multi-turn and evolution-driven process.

Reviewers MHr3 and 68ro view the contribution positively and mainly raised concerns about clarity, statistical metric, sensitivity analysis and fairness of comparisons. Reviewers 5QHQ and znBA mainly raise conceptual and framing concerns,  i.e., questioning whether “self-evolution” goes beyond standard RL, whether the hierarchical reward is over-engineered, and whether the contribution is fundamentally novel or mainly applied. The authors’ rebuttal provides detailed clarifications, additional ablations and agent-level behavioral evidence that strengthen the paper, but cannot fully overcome these reviewers’ philosophical disagreement with the terminology and positioning.

Overall, the work presents a good agentic RL approach with empirical merit, and the remaining disagreements reflect differences in interpretation of novelty and framing rather than deficiencies in methodology or experimental support. I am inclined to accepting this paper if the authors can reflect those key points raised by reviewers.

**Reviewer Concerns:**

I believe most of concerns of Reviewers MHr3 and 68ro are addressed. While I think reviewer 5QHQ  will still have concerns about the designs of RL and comparisons with other LLM baselines, I believe this paper has been improved a lot and meets the standard of acceptance.

**Reviewer Scores:**

I would say the final scores would be changed from 6,2,6,2 to 6, 4, 6, 6, since most of the concerns have been addressed, while for reviewer 5QHQ, they are partially addressed.

---

### Decision · Program_Chairs · 2026-01-26

Accept (Poster)